# TSDS: Data Selection for Task-Specific Model Finetuning

**Zifan Liu**
University of Wisconsin-Madison
Madison, WI
zliu676@wisc.edu

**Amin Karbasi**
Yale University
New Haven, CT
amin.karbasi@yale.edu

**Theodoros Rekatsinas**
Apple
Zürich, Switzerland
trekatsinas@apple.com

## Abstract

Finetuning foundation models for specific tasks is an emerging paradigm in modern machine learning. The efficacy of task-specific finetuning largely depends on the selection of appropriate training data. We present TSDS (**T**ask-**S**pecific **D**ata **S**election), a framework to select data for task-specific model finetuning, guided by a small but representative set of examples from the target task. To do so, we formulate data selection for task-specific finetuning as an optimization problem with a distribution alignment loss based on optimal transport to capture the discrepancy between the selected data and the target distribution. In addition, we add a regularizer to encourage the diversity of the selected data and incorporate kernel density estimation into the regularizer to reduce the negative effects of near-duplicates among the candidate data. We connect our optimization problem to nearest neighbor search and design efficient algorithms to compute the optimal solution based on approximate nearest neighbor search techniques. We evaluate our method on data selection for both continued pretraining and instruction tuning of language models. We show that instruction tuning using data selected by our method with a 1% selection ratio often outperforms using the full dataset and beats the baseline selection methods by 1.5 points in F1 score on average. Our code is available at https://github.com/ZifanL/TSDS.

## 1 Introduction

Finetuning foundation models [3] is the de-facto paradigm for building machine learning applications that focus on specific tasks. Models such as BERT [10] and LLaMA [43] are large-scale models pretrained on massive unlabeled data across a wide range of domains. Those models can be specialized to downstream tasks through finetuning. Finetuning can take a variety of forms depending on the target task. For instance, *continued pretraining* [17] extends the pretraining stage of a model on a dataset that is more closely related to a target domain. As another setting, *instruction tuning* [51] trains a generative foundation model on instruction-response pairs to improve its performance in responding to task-specific instructions.

Finetuning foundation models can lead to significant improvement in downstream tasks, but the effectiveness heavily relies on the right choice of training data [17, 30, 48, 47]. However, the data repositories that one considers during training of generative models tend to be large—consider for

example the use of Common Crawl[1], which contains 250 billion web pages, or The Pile [14]—and hence, it is impractical to manually select the data that are distributed like the use cases in the target task. Therefore, automated task-specific data selection becomes critical.

In this paper, we propose TSDS (**T**ask-**S**pecific **D**ata **S**election), a framework to select data for task-specific model finetuning. We consider the scenario of finetuning a foundation model to customize it for a specific task characterized by a few representative examples. The input to our framework is the representative examples and a massive repository of candidate data. Guided by the representative examples, we select training data from the repository for task-specific finetuning. We identify the following requirements for our framework.

(**Distribution Alignment**) First, the distribution of the selected data should match the distribution of the representative examples from the target task. Distribution alignment is essential for a model to learn the target distribution and enable data-efficient finetuning for the target task [40]. Many works [38, 17, 2, 50, 47] retrieve candidate examples that are most similar to the representative examples. Such heuristics do not ensure distribution alignment between the selected data and the representative examples. A recent work [48] selects data by importance resampling to match the target distribution but is limited to an n-gram feature space, which cannot capture high-level semantics.

(**Diversity**) Second, the selected data should be diverse so that the model can learn a wide range of related knowledge rather than overfitting to specific examples. In practice, data repositories created by web crawling may contain a large portion of near-duplicates [13, 28] that can compromise diversity and negatively impact model performance [28, 19]. For example, a study [13] on several snapshots of ClueWeb[2] and Common Crawl shows that 14% to 52% of the documents are near-duplicates. Previous works [38, 17, 2, 50, 48, 47] on task-specific data selection overlook near-duplicates, leading to the over-representation of such examples in the selected data. We require our framework to ensure diversity in selection even when a large fraction of the candidate examples are near-duplicates.

(**Scalability**) Finally, the selection algorithm should be efficient, considering the increasing scale of modern data repositories. The high volume of candidate data (e.g., 250 billion pages in Common Crawl) poses a great challenge to efficient selection.

Our framework formulates task-specific data selection as an optimization problem that allows a smooth trade-off between two crucial objectives: distribution alignment and diversity. The solution to the optimization problem is a categorical distribution assigned to the candidates which we will sample from. In the optimization objective, we use optimal transport to measure the discrepancy between the distribution assigned to the candidates and the target distribution, encouraging the alignment between them. We show that the optimization problem admits efficient algorithms to compute the optimal solution. In addition, our framework supports distribution alignment in any metric space that supports efficient nearest-neighbor search, including model-agnostic semantic embedding and model-specific features such as gradients.

Our contributions: 1) We formulate data selection for task-specific finetuning as an optimization problem based on optimal transport for distribution alignment, with a regularization term that encourages diversity. 2) We make our framework robust to near-duplicates by incorporating kernel density estimation [36] into the regularization term. 3)We show the connection between the optimal solution to the optimization problem and nearest neighbor search, which allows us to develop efficient algorithms employing approximate nearest-neighbor search techniques [23].

We conduct extensive experiments to validate the effectiveness of our framework. We focus on natural language processing tasks where foundation models have shown great advancements. We show that our framework beats the state-of-the-art baseline [47] by 1.5 points in F1 score on average with a selection ratio of 1% on instruction tuning for two modern large language models on three tasks. In addition, continued pretraining using domain-specific data selected by our framework outperforms the other selection methods by up to 3 F1 points on four classification tasks from various domains. We also demonstrate that our framework is robust to near-duplicates in the data repository, maintaining consistent performance when 1% of the candidate examples are duplicate for up to 1,000 times. Our method is efficient, taking 28 hours to preprocess a corpus of 150M examples and less than 1 hour for each task-specific selection.

---

[1]https://commoncrawl.org/
[2]https://lemurproject.org/

## 2 Background and Overview

In this section, we provide background information that is essential for the problem, followed by a formal statement of the problem and an overview of our proposed framework.

### 2.1 Background

We introduce the notations that will be used throughout the paper and the optimal transport problem.

**Notation** We use $\mathbb{R}_{\geq 0}$ to represent the set of non-negative real numbers, and $\mathbb{R}_{>0}$ to represent the set of positive real numbers. Let $N$ be a positive integer and we use $[N]$ to denote the set of integers from 1 to $N$. We use bold letters to denote matrices and the corresponding plain letters with subscripts to denote the entries in the matrix. For example, $\boldsymbol{\gamma} \in \mathbb{R}^{M \times N}$ is a matrix with size $M \times N$, and $\gamma_{ij}$ or $\gamma_{i,j}$ is the entry in the $i^{\text{th}}$ row and the $j^{\text{th}}$ column (1-indexed).

**Optimal Transport between Discrete Distributions** We introduce the optimal transport problem, which forms the basis of our data selection framework. Let $(A, f)$ be a metric space where $A$ is a finite set and $f : A \times A \to \mathbb{R}_{\geq 0}$ is a distance function. Consider two discrete distributions $\mu$ on $U \subseteq A$ and $\nu$ on $V \subseteq A$, where both $U$ and $V$ are finite sets. Let $u_i$ be the $i^{\text{th}}$ example in $U$ and $\mu_i = \mu(u_i)$ be the probability of $u_i$. Similarly, let $v_j$ be the $j^{\text{th}}$ example in $V$ and $\nu_j = \nu(v_j)$ be the probability of $v_j$. Let $\boldsymbol{\gamma} \in \mathbb{R}_{\geq 0}^{|U| \times |V|}$ be a transport of probability mass between $\mu$ and $\nu$, where $\gamma_{ij}$ is amount of probability mass transported from $u_i$ to $v_j$. Assume that the cost of transporting one unit of probability mass from $u_i$ to $v_j$ is $f(u_i, v_j)$, the distance between $u_i$ and $v_j$. Optimal transport is the problem of transporting all the probability mass from $U$ to $V$ with a minimal cost:

$$\min_{\boldsymbol{\gamma} \in \mathbb{R}_{\geq 0}^{|U| \times |V|}} \sum_{i=1}^{|U|} \sum_{j=1}^{|V|} \gamma_{ij} f(u_i, v_j) \quad \text{subject to} \quad \sum_{j=1}^{|V|} \gamma_{ij} = \mu_i, \forall i \in [|U|], \sum_{i=1}^{|U|} \gamma_{ij} = \nu_j, \forall j \in [|V|]$$

### 2.2 Task-Specific Data Selection Problem Statement

We now introduce the problem of data selection for task-specific finetuning. We assume access to a set of $M$ representative examples $Q = \{q_i\}_{i=1}^{M}$ from the target task, which we call query examples. Consider a data repository $D = \{x_j\}_{j=1}^{N}$ containing $N$ candidate examples. Note that $Q$ and $D$ are multisets that may contain duplicates. We aim to select $B$ examples from the repository guided by the query examples. The selected examples will be used to finetune a model to tailor it to the target task. We adopt the *model-agnostic* formulation above for the generality of the solution. However, our framework can be applied to model-specific selection by using model-specific data representations; an example evaluation for model-specific instruction tuning is presented in Section 5.1.

### 2.3 Framework Overview

Our framework takes the candidate examples and the query examples as inputs and outputs a set of task-specific examples by the following workflow. 1. (*Encoding*) We first encode the query examples and the candidate examples into the same metric space with a specified distance function. 2. (*Probability Assignment*) We determine the probability mass assigned to each candidate example by solving an optimization problem. 3. (*Sampling*) We take a random sample with replacement from the candidate examples following a categorical distribution where the probability is determined by the assignment in the previous step.

## 3 Data Selection and Optimal Transport

Data selection for task-specific finetuning can be expressed as an optimization problem for probability assignment to the candidates in the data repository. First, we discuss the formulation of the optimization problem and then show the existence of closed-form solutions. In addition, we propose a regularization term that addresses the problem of near-duplicates among the candidates. The proofs of the theorems in this section are provided in Appendix B.

## 3.1 Optimization Problem

Consider the metric space $(Z, f)$ where $Z = Q \cup D$ contains all the examples and $f : Z \times Z \to \mathbb{R}$ is a distance function. Let $\boldsymbol{d} \in \mathbb{R}_{\geq 0}^{M \times N}$ be the distance matrix, where $d_{ij} = f(q_i, x_j)$ is the distance between the $i^{\text{th}}$ query example and the $j^{\text{th}}$ candidate example.

We propose an optimization problem that transports probability mass from the query examples to the candidates. The objective is a linear combination of a probability transport cost for distribution alignment and a regularization term to encourage diversity. Formally, given $\boldsymbol{d} \in \mathbb{R}_{\geq 0}^{M \times N}$, we consider the following optimization problem, which we refer to as Problem RT (regularized transport):

$$\min_{\boldsymbol{\gamma} \in \mathbb{R}_{\geq 0}^{M \times N}} \frac{\alpha}{C} \sum_{i=1}^{M} \sum_{j=1}^{N} \gamma_{ij} d_{ij} + (1 - \alpha) G(\boldsymbol{\gamma}) \quad \text{subject to} \quad \sum_{j=1}^{N} \gamma_{ij} = \frac{1}{M}, \forall i \in [M]$$

where $C > 0$ is a scaling constant, $\alpha \in [0, 1]$ is a hyper-parameter that controls the trade-off between distribution alignment and diversity, and $G$ is a regularization function. The first term in Problem RT is the cost of probability transport where $\gamma_{ij}$ is the mass transported from the $i^{\text{th}}$ query example to the $j^{\text{th}}$ candidate. Each query example has $\frac{1}{M}$ probability mass to transport, as stated in the constraint. The probability transport cost measures the cost of transforming one distribution to another by moving probability mass between them, providing a method to quantify probability alignment. The second is a regularization term that encourages the diversity of probability transport.

Let $\boldsymbol{\gamma}^*$ be an optimal solution to Problem RT. We assign $p_j^* = \sum_{i=1}^{M} \gamma_{ij}^*$ probability to candidate example $x_j$, which is the sum of the probability mass it receives from all the query examples. When we sample from the candidate examples in the subsequent step, $x_j$ has probability $p_j^*$.

We propose two instantiations of the regularization term that encourage the diversity of probability transport by penalizing its discrepancy to the uniform transport:

- $G_\infty(\boldsymbol{\gamma}) = M \max_{i \in M, j \in N} |\gamma_{ij} - \frac{1}{MN}|$ captures the largest probability gap between $\boldsymbol{\gamma}$ and the uniform transport.

- $G_{\text{TV}}(\boldsymbol{\gamma}) = \frac{1}{2} \sum_{i=1}^{M} \sum_{j=1}^{N} |\gamma_{ij} - \frac{1}{MN}|$ is the total variation distance between $\boldsymbol{\gamma}$ and the uniform transport.

We use uniform transport as a reference point to encourage diversity as it represents the most diverse way of transporting the probability mass from one query example to all the candidates, assuming the candidates are distinct.

## 3.2 Closed-Form Solution

When $G = G_\infty$, Problem RT can be solved by standard linear programming techniques, but they run in $\Omega((MN)^2)$ time, which is prohibitively expensive. Instead, we show the existence of a closed-form solution that can be computed in $O(MN \log N)$ time (see Section 4 for the algorithm).

Using $G_\infty$ as the regularization function, we get an optimal solution by transporting the probability of each query example evenly to its $K$-nearest neighbors among the candidates, where $K$ is determined by the tradeoff between distribution alignment and diversity:

**Theorem 3.1.** *Given $\boldsymbol{d} \in \mathbb{R}_{\geq 0}^{M \times N}$ where $N > 1$, consider Problem RT with $G(\boldsymbol{\gamma}) = G_\infty(\boldsymbol{\gamma}) = M \max_{i \in M, j \in N} |\gamma_{ij} - \frac{1}{MN}|$. For all $i \in [M]$, let $j_1^i, \ldots, j_N^i$ be a reordering of $[N]$ such that $d_{ij_1^i} \leq \cdots \leq d_{ij_N^i}$. Consider $\boldsymbol{\gamma}^* \in \mathbb{R}_{\geq 0}^{M \times N}$ whose entries are $\frac{1}{KM}$ if $j \in \{j_1^i, \ldots, j_K^i\}$ and 0 otherwise, where $K = \max\{k \in [N] | \frac{\alpha}{C} \sum_{i=1}^{M} \sum_{l=1}^{k-1} (d_{ij_k^i} - d_{ij_l^i}) < (1 - \alpha)M\}$. Assume $K \leq N/2$, and then $\boldsymbol{\gamma}^*$ is a minimizer of Problem RT. $\boldsymbol{\gamma}^*$ is the unique minimizer if $\frac{\alpha}{C} \sum_{i=1}^{M} \sum_{l=1}^{K} (d_{ij_{K+1}^i} - d_{ij_l^i}) > (1 - \alpha)M$ and $\nexists i \in [M]$ such that $d_{ij_{K+1}^i} = d_{ij_K^i}$.*

Similarly, there exists a closed-form solution that can be computed in $O(MN \log N)$ time when $G = G_{\text{TV}}$ (see Appendix A for the solution and the algorithm).

### 3.3 Addressing Near-Duplicates via Kernel Density Estimation

When there exists a large fraction of near-duplicates among the candidates, $G_\infty$ fails to characterize the diversity of probability assignment since it treats near-duplicates as distinct examples. Consequently, the contents in the near-duplicates will be over-sampled. For example, if 100 of the $K$-nearest neighbors of a query example are duplicates and the others are distinct, the content in the duplicates will receive 100 times as much probability mass as any other example.

To address the near-duplicate problem, we propose a regularization function incorporating kernel density estimation (KDE) [36], which is a non-parametric method to estimate the probability density function from finite examples. We determine the duplication level of a point by the kernel density estimate at its position. We use the Epanechnikov kernel such that given $D$, the density estimate at point $x$ is $\sum_{x' \in D} \max(1 - \frac{f(x,x')^2}{h^2}, 0)$, where $h > 0$ is the kernel size and $f$ is the distance function. For example, for a point $x$ in $D$ whose distance to any other point is larger than $h$, the density estimate at $x$ is 1. If we create two duplicates of $x$ and add them to $D$, the density estimate at $x$ increases to 3.

Our KDE-based regularization function is $G_{\text{KDE}}(\gamma) = M \max_{i \in [M], j \in [N]} \rho_j |\gamma_{ij} - \frac{1/\rho_j}{M \sum_{j' \in [N]} 1/\rho_{j'}}|$ where $\rho_j = \sum_{x' \in D}(1 - f(x_j, x')/h^2)$ is the density estimate at $x_j$. $G_{\text{KDE}}(\gamma)$ compares $\gamma$ to the probability assignment that is proportional to the inverse of the density, and penalizes the largest gap weighted by the density. Note that $G_\infty$ is a special case of $G_{\text{KDE}}(\gamma)$ with $\rho_j = 1$ for all $j \in [N]$.

The optimal solution to Problem RT when $G = G_{\text{KDE}}$ can be obtained by assigning the probability mass of each query example to the nearest neighbors among the candidates, weighted by the inverse of their density estimate, as is shown by the following theorem.

**Theorem 3.2.** *Given $d \in \mathbb{R}_{\geq 0}^{M \times N}$ and $\rho_1, \ldots, \rho_N \in \mathbb{R}_{>0}$, consider Problem RT with $G(\gamma) = G_{KDE}(\gamma) = M \max_{i \in [M], j \in [N]} \rho_j |\gamma_{ij} - \frac{1/\rho_j}{M \sum_{j' \in [N]} 1/\rho_{j'}}|$. For all $i \in [M]$, let $j_1^i, \ldots, j_N^i$ be a reordering of $[N]$ such that $d_{ij_1^i} \leq \cdots \leq d_{ij_N^i}$. Let $s_k^i = \sum_{l=1}^{k} 1/\rho_{j_l^i}$, and $s$ be a discrete variable that takes value from $\mathcal{S} = \{s_k^i | i \in [M], k \in [N]\} \cup \{0\}$. Let $c(s) = \sum_{i=1}^{M} c_i(s)$, where $c_i(s) = 0$ if $s \leq s_1^i$ and $c_i(s) = \sum_{l=1}^{k-1} \frac{d_{ij_k^i} - d_{ij_l^i}}{\rho_{j_l^i}}$ if $s_{k-1}^i < s \leq s_k^i$ for any $k \geq 2$. Let $s^* = \max\{s \in \mathcal{S} | \frac{\alpha}{C} c(s) < (1 - \alpha)M\}$, and $K_i = \max\{k \in \{0, \ldots, N-1\} | s_k^i \leq s^*\}$. Assume $s^* \leq \frac{1}{2} \sum_{j=1}^{N} 1/\rho_j$, and then $\gamma^*$ is a minimizer of Problem RT where $\forall i \in [M], k \in [N]$*

$$\gamma_{ij_k^i}^* = \begin{cases} 1/(Ms^* \cdot \rho_{j_k^i}), & \text{if } k \leq K_i \\ \frac{1}{M} - \sum_{l=1}^{K_j} 1/(Ms^* \cdot \rho_{j_l^i}), & \text{if } k = K_i + 1 \\ 0, & \text{otherwise} \end{cases}$$

*$\gamma^*$ is the unique minimizer if $\nexists s \in \mathcal{S}$ such that $\frac{\alpha}{C} c(s) = (1 - \alpha)M$ and $\nexists i \in [M]$ such that $d_{ij_{K_i}^i} = d_{ij_{K_i+1}^i}$ or $d_{ij_{K_i+1}^i} = d_{ij_{K_i+2}^i}$.*

Intuitively, we count candidate $x_j$ as $1/\rho_j$ examples. For each query example, the optimal solution assigns probability mass to the candidates in its neighborhood proportional to their adjusted counts. The size of the neighborhood is determined by the limit $s^*$ on the sum of the adjusted counts.

In Figure 1, we show an example comparing the optimal transport with $G_\infty$ and $G_{\text{KDE}}$. When $G = G_\infty$, the probability is transported uniformly to the candidates regardless of their relative positions. When $G = G_{\text{KDE}}$, the clustered candidates receive less probability due to their high density, and they will be less over-represented when we take samples according to the assigned probability.

## 4 Efficient Probability Assignment Algorithms for Data Selection

We propose efficient algorithms to assign probability mass to the candidates according to the optimal solutions to Problem RT. For $G = G_\infty$ and $G = G_{\text{KDE}}$, the corresponding algorithms are KNN-Uniform (Algorithm 1) and KNN-KDE (Algorithm 2). Each algorithm takes the query examples and the candidates as input and outputs the probability assigned to each candidate.

Both algorithms prefetch the $L$ nearest neighbors of each query example from the candidates as the first step, where $L$ is a limit on the neighborhood size. Specifically, GETKNN$(\mathcal{Q}, \mathcal{D}, L)$ returns the

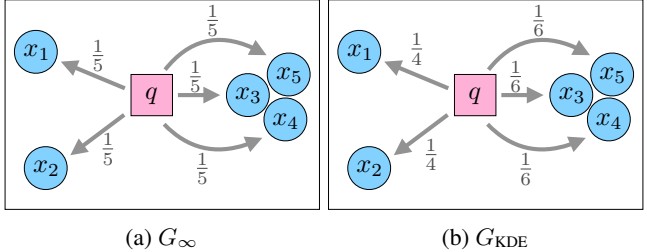

(a) $G_\infty$  (b) $G_{\text{KDE}}$

Figure 1: An example of the optimal probability transports under different regularization terms. We consider 1 query example $q$ and 5 candidates $x_1, \ldots, x_5$ embedded in a 2-dimensional space. Assume that the candidates that form a cluster (i.e., $x_3, x_4, x_5$) have a density estimate of $\frac{3}{2}$ each and the others have a density estimate of 1.

---

**Algorithm 1:** KNN-Uniform.

---

1   **Input:** query examples $\mathcal{Q} = \{q_i\}_{i=1}^M$, candidates $\mathcal{D} = \{x_j\}_{j=1}^N$, number of nearest neighbors to prefetch $L$, $\alpha \in [0, 1]$, $C > 0$; **Output:** $p_1, \ldots, p_N$;

2   $\boldsymbol{j}, \boldsymbol{d} \leftarrow \text{GETKNN}(\mathcal{Q}, \mathcal{D}, L); K \leftarrow 1$;

3   **while** $K < L$ *and* $\frac{\alpha}{C} \sum_{i=1}^M \sum_{k=1}^K [d_{i,K+1} - d_{ik}] < (1 - \alpha)M$ **do**

4     $\mid$   $K \leftarrow K + 1$;

5   **foreach** $j \in [N]$ **do**

6     $\mid$   $p_j \leftarrow 0$;

7   **foreach** $i \in [M]$ **do**

8     **foreach** $k \in [K]$ **do**

9       $\mid$   $p_{j_{ik}} \leftarrow p_{j_{ik}} + \frac{1}{KM}$;

---

indices $\boldsymbol{j} \in \mathbb{N}^{M \times L}$ of the nearest neighbors and the corresponding distances $\boldsymbol{d} \in \mathbb{R}^{M \times L}$, where $j_{ik}$ is the index of the $k^{\text{th}}$ nearest neighbor of $q_i$ in $\mathcal{D}$, and $d_{ik}$ is the distance between $q_i$ and $x_{j_{ik}}$. Retrieving nearest neighbors exactly requires computing the distance between every query example and all the candidates, which is inefficient when the candidate size $N$ is in the order of millions and billions. Alternatively, we can employ approximate nearest search techniques [23, 16] to improve efficiency at the cost of lower accuracy.

Then the algorithms assign probability mass to the nearest neighbors of each example. KNN-Uniform determines $K$ based on the tradeoff between distribution alignment and diversity. Then the algorithm assigns the probability mass of each query example evenly to its $K$-nearest neighbors. KNN-KDE assigns probability mass to the nearest neighbors proportional to the inverse of their kernel density estimates (Line 15-18). The sizes of the neighborhoods are determined by Line 7-12, where we increase the limit $s$ on the sum of the inverse of the density estimates over the neighborhood until the condition on Line 9 is satisfied. We use a priority queue to store the possible values $s$ can take and retrieve the smallest one in each iteration.

In KDE-KNN, we also precompute the kernel density estimate for the $L$-nearest neighbors of each query example. To estimate the kernel density of each candidate example, we need to compute the distance between it and all the other candidate examples. To reduce the computational cost, we use the $I$-nearest neighbors among the prefetched examples as the set to compute KDE for each candidate example. Let $\mathcal{D}'$ be the set containing the $L$-nearest neighbors of all the query points and $\mathcal{N}_x$ be the $I$-nearest neighbors of $x$ in $\mathcal{D}'$. We compute the KDE of example $x$ as $\sum_{x' \in \mathcal{N}_x} (1 - \frac{f(x,x')^2}{h^2})$.

KNN-Uniform runs in $O(ML + T_1)$ time, and KNN-KDE runs in $O(ML \log M + T_2)$ time, where $T_1$ is the runtime of GETKNN, and $T_2$ is the runtime of COMPUTEKDE. With exact nearest neighbor search, $T_1 = O(MN \log N)$ and $T_2 = O(M^2 L^2 \log(ML))$. If we employ approximate nearest neighbor search techniques such as HNSW [34] for real vectors and $l_2$ distance, we have $T_1 = O((M + N) \log N)$ and $T_2 = O(ML \log(ML))$.

**Algorithm 2:** KNN-KDE.

---

1 **Input:** query examples $\mathcal{Q} = \{q_i\}_{i=1}^M$, candidate examples $\mathcal{D} = \{x_j\}_{j=1}^N$, number of nearest neighbors to prefetch $L > 1$, $\alpha \in [0,1]$, $C > 0$; **Output:** $p_1, \ldots, p_N$;

2 $\boldsymbol{j}, \boldsymbol{d} \leftarrow \text{GETKNN}(\mathcal{Q}, \mathcal{D}, L)$;

3 $\boldsymbol{\rho} \leftarrow \text{COMPUTEKDE}(\boldsymbol{j}, \mathcal{D})$ /* $\boldsymbol{\rho} \in \mathbb{R}^{M \times L}$ and $\rho_{ik}$ is the density of $x_{j_{ik}}$   */

4 $\mathcal{H} \leftarrow \text{EmptyPriorityQueue}()$;

5 **for** $i \in [M]$ **do**

6    $K_i \leftarrow 0$; $c_i \leftarrow 0$; $\mathcal{H}.\text{push}((1/\rho_{i1}, i))$;

7 **while** $\mathcal{H}$ *is not empty* **do**

8    $s, i \leftarrow \mathcal{H}.\text{pop}()$; $K_i \leftarrow K_i + 1$; $c_i \leftarrow \sum_{k=1}^{K_i}(d_{i,K_i+1} - d_{ik})/\rho_{ik}$;

9    **if** $\frac{\alpha}{C}\sum_{i=1}^M c_i \geq (1-\alpha)M$ **then**

10       $s^* \leftarrow s$; break;

11    **if** $K_i + 1 < L$ **then**

12       $\mathcal{H}.\text{push}((s + 1/\rho_{i,K_i+1}, i))$;

13 **for** $j \in [N]$ **do**

14    $p_j \leftarrow 0$;

15 **for** $i \in [M]$ **do**

16    **for** $k \in [K_i]$ **do**

17       $p_{j_{ik}} \leftarrow p_{j_{ik}} + 1/(Ms^* \cdot \rho_{ik})$;

18    $p_{j_{i,K_i+1}} \leftarrow p_{j_{i,K_i+1}} + \frac{1}{M} - \sum_{k=1}^{K_i} 1/(Ms^* \cdot \rho_{ik})$;

---

Table 1: Information of the target datasets for instruction tuning.

| Dataset | Task | # Test Instances | # Query Examples | # Shots* | Metric |
|---------|------|------------------|------------------|----------|--------|
| TydiQA [7] | Multilingual QA | 1,713 | 9 | 1 | F1 score |
| MMLU [18] | Multiple choice | 18,721 | 285 | 5 | Accuracy |
| BBH [41] | Reasoning | 920 | 81 | 3 | Accuracy |

*# shots is the number of QA examples provided in the prompt when querying the model.

## 5 Experiments

We evaluate our framework on data selection for task-specific instruction tuning and domain-specific continued pretraining, using different encodings as needed. We show that 1) our framework outperforms the state-of-the-art methods on data selection for task-specific instruction tuning and domain-specific continued pretraining by up to 6 points and 3 points in F1 score respectively; 2) our framework is robust to duplicates, exhibiting consistent performance when 1% of the candidate examples are duplicated up to 1000 times, while baseline methods show a drop of 2 points in F1 score (see Appendix E.1); 3) our method is efficient, requiring 28 hours to preprocess 150 million candidate examples and less than 1 hour for each task-specific selection (see Appendix E.2).

### 5.1 Evaluation on Task-Specific Instruction Tuning

We select training data to perform instruction tuning to tailor a model to specific downstream tasks. We assume access to several query examples that represent the use cases of the target task and a repository of instruction-response pairs to select from. The detailed setting is as follows.

**Target Tasks, Model, and Data Repository** We consider three tasks from standard benchmarks for language model evaluation. The properties are shown in Table 1. We use two models: LLAMA-2-7B [43] and MISTRAL-7B [22]. We use a combination of FLAN V2 [31], COT [45], DOLLY [8], and OPEN ASSISTANT [26] as the data repository for selection, which contains 270K examples.

**Encoding** We encode the examples using rescaled and randomly projected gradients from a LLAMA-2-7B model finetuned on a random 5% of the data repository. The encoding process follows Xia et al. [47], who show that gradient-based encoding is essential to capture the utility of training examples in instruction tuning. We use $l_2$ distance as the distance function. See Appendix C for the details.

**Methods**   1) **Rand** selects a random subset from the data repository; 2) **LESS** [47] (the state-of-the-art method on data selection for task-specific instruction tuning) selects training data from the data repository based on their gradient similarity to the query examples; 3) **Ours** is the KNN-KDE instantiation of our framework with $C = 5$, $\alpha = 0.075$ and $h = 0.2$. We discuss how we choose the parameters in Appendix C. The implementation details of our method can also be found in Appendix C. Note that our method is not sensitive to the hyperparameters, as shown by the microbenchmarks in Appendix E.

**Evaluation Protocol**   Following Xia et al. [47], we finetune the base model on the selected data for 4 epochs. The dataset size is 0.5% / 1.0% / 5% of the data repository. Since our method is based on probabilistic sampling, we do not select a fixed training set. Instead, in each epoch we sample randomly from the data repository following the assigned probability. The hyperparameters for finetuning also follow Xia et al. [47] (see Appendix D). We repeat each experiment for three runs with different random seeds and report the mean and standard deviation.

Table 2: Performance of instruction tuning with dataset selected by our method compared with the baselines. The subscripts represent the standard deviations.

| Model | | LLAMA-2-7B | | | MISTRAL-7B | | |
|---|---|---|---|---|---|---|---|
| Dataset | | TydiQA | MMLU | BBH | TydiQA | MMLU | BBH |
| Base | | 40.6 | 45.7 | 39.1 | 49.6 | 62.4 | 56.5 |
| Full | | 52.7 | 51.4 | 41.4 | 44.7 | 58.9 | 48.0 |
| Ratio 0.5% | Rand | $49.8_{2.4}$ | $45.0_{0.4}$ | $38.3_{0.5}$ | $57.0_{1.5}$ | $59.5_{0.3}$ | $49.7_{0.1}$ |
| | LESS | $52.3_{1.4}$ | $46.2_{0.7}$ | $39.0_{0.6}$ | $55.0_{3.0}$ | $\mathbf{60.6_{0.5}}$ | $53.0_{0.9}$ |
| | Ours | $\mathbf{53.7_{1.5}}$ | $\mathbf{47.2_{0.2}}$ | $\mathbf{40.6_{0.2}}$ | $\mathbf{61.6_{0.9}}$ | $60.3_{0.9}$ | $\mathbf{55.0_{1.7}}$ |
| Ratio 1.0% | Rand | $47.8_{1.7}$ | $45.9_{0.5}$ | $38.2_{0.7}$ | $57.8_{0.4}$ | $59.4_{0.2}$ | $53.7_{1.0}$ |
| | LESS | $54.0_{1.0}$ | $\mathbf{48.3_{0.2}}$ | $40.2_{0.6}$ | $59.0_{0.8}$ | $\mathbf{61.1_{0.1}}$ | $53.7_{1.8}$ |
| | Ours | $\mathbf{55.4_{0.5}}$ | $47.9_{0.2}$ | $\mathbf{42.0_{1.1}}$ | $\mathbf{63.6_{1.4}}$ | $60.5_{0.8}$ | $\mathbf{56.3_{2.1}}$ |
| Ratio 5.0% | Rand | $49.5_{1.4}$ | $46.0_{0.8}$ | $40.8_{0.6}$ | $57.6_{0.7}$ | $60.2_{0.3}$ | $54.8_{1.1}$ |
| | LESS | $54.3_{0.7}$ | $50.6_{0.0}$ | $40.2_{1.8}$ | $60.4_{1.3}$ | $\mathbf{61.3_{0.5}}$ | $53.7_{0.6}$ |
| | Ours | $\mathbf{54.3_{1.0}}$ | $\mathbf{50.9_{0.4}}$ | $\mathbf{42.7_{0.2}}$ | $60.9_{1.8}$ | $59.9_{0.4}$ | $\mathbf{56.0_{0.5}}$ |

**Results**   The results are shown in Table 2 where "Base" is the base model without finetuning and "Full" is the model finetuned on the full data repository. Our method consistently outperforms the baselines on TydiQA and BBH across different selection ratios, beating the state-of-the-art method (LESS) by up to 6 points. With a selection ratio of 1%, our method outperforms the full data repository on TydiQA and BBH. On MMLU, our methods show comparable results to LESS. Note that for MISTRAL-7B, finetuning on the full repository leads to worse performance than no finetuning, which highlights the importance of careful data selection for task-specific instruction tuning. We also notice that finetuning MISTRAL-7B on any selected set does not increase its accuracy on MMLU. The reason could be that the base MISTRAL-7B model has already been well-tuned for multiple-choice questions using high-quality data. We observe a drop in the performance of our method on TydiQA when the selection ratio increases from 1% to 5%, which may be caused by overfitting. We can early stop the training process to avoid overfitting in practice.

## 5.2   Evaluation on Domain-Specific Continued Pretraining

In this experiment, we select data for domain-specific continued pretraining to adapt a model to a specific domain. We assume access to a set of annotated data for a domain-specific task that serves as query examples and a repository of unlabeled data to select from. We continue pretraining the base model on the selected data and then perform supervised finetuning using the annotated data.

**Target Tasks and Data Repository**   We consider four datasets focused on classification tasks across diverse domains. The properties are provided in Table 3. We select data for continued pertaining from a data repository consisting of 150M sequences crafted by Xie et al. [48] from The Pile [14].

Table 3: Training, validation, test sizes and the number of classes in the datasets.

| Dataset | Domain | Train | Validation | Test | Classes | Metric |
|---------|--------|-------|-----------|------|---------|--------|
| ChemProt [25] | Biomedical | 4,169 | 2,427 | 3,469 | 13 | micro-F1 score |
| IMDB [33] | Movie review | 20,000 | 5,000 | 25,000 | 2 | macro-F1 score |
| SCIERC [32] | Computer science | 3,219 | 455 | 974 | 7 | macro-F1 score |
| AGNews [52] | News | 114,947 | 4,999 | 7,596 | 4 | macro-F1 score |

Table 4: F1 scores of the downstream tasks. Standard deviations are shown in the subscripts.

| | ——1K Annotated Data—— | | | | ——3K Annotated Data—— | | | | 10K Annotated Data | |
|------|--------|--------|--------|--------|--------|--------|--------|--------|--------|--------|
| | ChemP. | IMDB | SCI. | AGNews | ChemP. | IMDB | SCI. | AGNews | IMDB | AGNews |
| Base | $69.6_{1.8}$ | $88.0_{0.4}$ | $60.1_{2.3}$ | $87.1_{0.2}$ | $77.1_{1.1}$ | $88.7_{0.4}$ | $75.8_{1.1}$ | $87.7_{0.3}$ | $90.0_{0.0}$ | $89.1_{0.1}$ |
| Rand | $69.7_{1.5}$ | $87.3_{0.1}$ | $62.7_{2.9}$ | $87.2_{0.3}$ | $78.6_{0.2}$ | $88.5_{0.1}$ | $77.5_{1.6}$ | $88.2_{0.1}$ | $90.2_{0.1}$ | $90.2_{0.1}$ |
| DSIR | $74.8_{0.7}$ | $87.7_{0.6}$ | $68.5_{0.1}$ | $\mathbf{87.4_{0.2}}$ | $\mathbf{82.2_{0.4}}$ | $89.4_{0.2}$ | $78.9_{0.7}$ | $89.1_{0.3}$ | $90.8_{0.1}$ | $90.1_{0.1}$ |
| Ours | $\mathbf{76.7_{0.6}}$ | $\mathbf{89.8_{0.1}}$ | $\mathbf{72.1_{0.6}}$ | $87.3_{0.2}$ | $81.9_{0.4}$ | $\mathbf{90.7_{0.0}}$ | $\mathbf{79.2_{0.9}}$ | $\mathbf{89.3_{0.1}}$ | $\mathbf{91.6_{0.1}}$ | $\mathbf{90.7_{0.1}}$ |

**Target-Domain Data Accessibility** To simulate different levels of access to target-domain annotated data, we consider three settings with varying sizes of annotated data (1K, 3K, and 10K). When the size is set to $M$ and the original target-domain training set is larger than $M$, we sub-sample it by choosing $M$ examples uniformly at random without replacement.

**Encoding** We encode the examples into $\mathbb{R}^{512}$ using the Universal Sentence Encoder [5] to capture semantic meanings and use $l_2$ distance as the distance function.

**Methods** 1) **Rand** selects a random subset from the data repository; 2) **DSIR** [47] (the state-of-the-art method on data selection for domain-specific continued pretraining) selects examples by importance resampling to match the unigram and bigram distribution of the query examples.; 3) **Ours** is the KNN-KDE instantiation of our framework with $C = 5$, $\alpha = 0.6$ and $h = 0.1$.

**Evaluation Protocol** For each domain-specific task, we provide the annotated set to the selection methods as the query examples to guide the selection. We perform continued pretraining on 1M examples selected by each method from the data repository for one epoch (see Appendix E.4 for different selection sizes), starting from the base ALBERT [27] model. Then we finetune the model on the domain-specific annotated set and evaluate it on the test set. The hyperparameters for training follow previous works [17, 49, 48] (see Appendix D). The experiments are repeated five times with varying random seeds. We remove the best and the worst among the five runs to rule out outlier runs and report the mean and standard deviation.

**Results** The test F1 scores of the downstream classification tasks are reported in Table 4. As a reference point, we provide the performance of finetuning the model directly without continued pretraining (Base). Our method outperforms the baselines in most cases except ChemProt (3K) and AGNews (1K), with a gap of up to 3 points in F1 scores. On ChemProt (3K) and AGNews (1K), our method is comparable to DSIR. We also notice that our method shows an average improvement of 1.92 points over DSIR with an annotated set size of 1K and 0.38 points with an annotated set size of 3K. This indicates that our method is particularly effective with small annotated sets.

## 6 Related Works

**Task-Specific Data Selection** Similarity-based methods [39, 17, 2, 50] retrieves the top ones from the candidates, ranked by their similarity to the representative examples from the target task. The features used for similarity computation can be embeddings or ngrams for texts. Another line of works [35, 48] use two generative models where one learns the distribution of the target-task data and the other learns the general-purpose data. Model-specific data selection methods [12, 47] choose data to maximize the model performance on the target task. Given the high cost of actually training a model and evaluating it on the target task, these methods often estimate the model performance by approximation. DSDM [12] approximate the model performance using datamodels [21], a function

that maps the training data membership (whether each candidate is included in the training set or not) to the model performance. LESS [47] employs the influence function [24] to approximate the marginal gain on the model performance when including a candidate into the training set. Specifically, LESS computes the gradient similarity between each candidate and all the query examples, and the maximum similarity is the score for ranking. Then the top-ranked candidates are selected. A major difference between our method and LESS is that our method matches the distributions, while LESS takes the top ones based on aggregated statistics.

**Diversity Measurement for Data Selection**  Measuring diversity is a critical aspect of data selection, as it ensures that the chosen dataset represents a wide range of examples rather than being overly concentrated on similar or redundant instances. DEITA [29] selects data in an iterative manner, where the contribution of a new example to the overall diversity is measured by the clipped cosine distance between the new example and the closest examples that have been selected. QDIT [4] measures the diversity of the selected data using the facility location function that quantifies how well each example in the full set is represented by the selected set. Wang et al. [44] measure the diversity using the log determinant distance between the selected set and a reference set that is maximally diverse.

**Data Deduplication**  Data deduplication removes duplicates or near-duplicates from a dataset. Exact duplicates can be detected using hash functions [11, 46], while the detection of near-duplicates is more challenging. Some works [37, 14] identify near-duplicates utilizing locality-sensitive hashing [15]. Others [28, 6] compute edit distances between examples to find near-duplicates. Another line of works [1, 42] relies on learned embeddings of the examples to detect near-duplicates.

## 7  Conclusion

In this paper, we proposed a framework for data selection for task-specific model finetuning, based on optimal transport, which allows a smooth tradeoff between distribution alignment and diversity. We incorporated kernel density estimation to make the selection robust to near-duplicates. Experimentally we showed that our method is effective in both task-specific instruction tuning and domain-specific continued pretraining. A potential direction for future work is to incorporate more efficient variants of optimal transport, such as Sinkhorn distances [9], to further improve the computational efficiency. One limitation of our framework is the reliance on a set of representative examples to guide the selection, which may not be easy to craft. The representative examples may also contain biases that can be exaggerated through the selection process, leading to negative social impacts. In practice, additional effort must be allocated to ensure the quality of the representative examples and the size of the representative examples needs to be decided according to the budget of human effort.

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

# A  Closed-Form Solution and Algorithm for $G_{\text{TV}}$

When $G = G_{\text{TV}}$, for each query example, we transport $\frac{1}{MN}$ probability mass to any candidate example whose distance to the query example is less than $\frac{(1-\alpha)C}{2\alpha}$ plus the distance between the query example and its 1-nearest neighbor. Then we transport all the remaining probability mass to the 1-nearest neighbor of each query example.

**Theorem A.1.** *Given $d \in \mathbb{R}_{\geq 0}^{M \times N}$ where $N > 1$, consider Problem RT with $G(\gamma) = G_{TV}(\gamma) = \frac{1}{2} \sum_{i=1}^{M} \sum_{j=1}^{N} |\gamma_{ij} - \frac{1}{MN}|$. For all $i \in [M]$, let $j_1^i, \ldots, j_N^i$ be a reordering of $[N]$ such that $d_{ij_1^i} \leq \cdots \leq d_{ij_N^i}$. Consider $\gamma^* \in \mathbb{R}_{\geq 0}^{M \times N}$ where $\forall i \in [M]$*

$$\forall k \in \{2, \ldots, N\}, \gamma_{ij_k^i}^* = \begin{cases} \frac{1}{MN}, & \text{if } d_{ij_k^i} - d_{ij_1^i} < \frac{(1-\alpha)C}{\alpha} \\ 0, & \text{otherwise} \end{cases}$$

*and*

$$\gamma_{ij_1^i}^* = \frac{1}{M} - \sum_{k=2}^{N} \gamma_{ij_k^i}^*$$

*Then $\gamma^*$ is a minimizer of Problem RT. $\gamma^*$ is the unique minimizer if $\forall i \in [M] \forall k \in [N], d_{ij_k^i} - d_{ij_1^i} \neq \frac{(1-\alpha)C}{\alpha}$ and $d_{ij_1^i} \neq d_{ij_2^i}$.*

The corresponding algorithm is KNN-T (Algorithm 3). KNN-TV assigns $\frac{1}{MN}$ unit of probability mass to the nearest neighbors that satisfy the distance condition in Line 9 and the rest to the 1-nearest neighbor. KNN-TV has the same time complexity as KNN-Uniform.

---

**Algorithm 3:** KNN-TV.

---

1 **Input:** query examples $\mathcal{Q} = \{q_i\}_{i=1}^{M}$, candidate examples $\mathcal{D} = \{x_j\}_{j=1}^{N}$, number of nearest neighbors to prefetch $L$, $\alpha \in [0, 1]$, $C > 0$;
2 **Output:** $p_1, \ldots, p_N$;
3 $j, d \leftarrow \text{GETKNN}(\mathcal{Q}, \mathcal{D}, L)$;
4 **for** $j \in [N]$ **do**
5 $\quad p_j \leftarrow 0$;
6 **for** $i \in [M]$ **do**
7 $\quad p_{j_{i1}} \leftarrow p_{j_{i1}} + \frac{1}{M}$;
8 $\quad k \leftarrow 2$;
9 $\quad$ **while** $k \leq L$ *and* $\frac{\alpha}{C}(d_{ik} - d_{i1}) < \frac{1}{2}(1 - \alpha)$ **do**
10 $\quad\quad p_{j_{ik}} \leftarrow p_{j_{ik}} + \frac{1}{MN}$;
11 $\quad\quad p_{j_{i1}} \leftarrow p_{j_{i1}} - \frac{1}{MN}$;
12 $\quad\quad k \leftarrow k + 1$;

---

# B  Proofs

## B.1  Proof of Theorem A.1

*Proof.* Let $\mathcal{L}(\gamma) = \frac{\alpha}{C} \sum_{i=1}^{M} \sum_{j=1}^{N} \gamma_{ij} d_{ij} + (1 - \alpha) G_{\text{TV}}(\gamma)$ be the optimization objective. We prove the theorem by showing that for any $\gamma' \in \mathbb{R}_{\geq 0}^{M \times N}$ that satisfy the constraint $(\forall i \in [M] \sum_{j=1}^{N} \gamma'_{ij} = \frac{1}{M})$, $\mathcal{L}(\gamma') \geq \mathcal{L}(\gamma^*)$.

Let $\gamma'' \in \mathbb{R}_{\geq 0}^{M \times N}$ be a probability transport such that

$$\forall k \in \{2, \ldots, N\}, \gamma''_{ij_k^i} = \begin{cases} \gamma'_{ij_k^i}, & \text{if } d_{ij_k^i} - d_{ij_1^i} < \frac{(1-\alpha)C}{\alpha} \\ 0, & \text{otherwise} \end{cases}$$

We show that $\mathcal{L}(\boldsymbol{\gamma}') \geq \mathcal{L}(\boldsymbol{\gamma}'')$. For any $i \in [M]$, let $\hat{k}_i = \max\{k \in [N] | d_{ij_k^i} - d_{ij_1^i} < \frac{(1-\alpha)C}{\alpha}\}$. Then we have

$$
\begin{aligned}
\mathcal{L}(\boldsymbol{\gamma}') - \mathcal{L}(\boldsymbol{\gamma}'') =& \frac{\alpha}{C}\sum_{i=1}^{M}\sum_{j=1}^{N}(\gamma_{ij}' - \gamma_{ij}'')d_{ij} + \frac{1-\alpha}{2}\sum_{i=1}^{M}\sum_{j=1}^{N}(|\gamma_{ij}' - \frac{1}{MN}| - |\gamma_{ij}'' - \frac{1}{MN}|) \\
=& \sum_{i=1}^{M}\sum_{j=1}^{N}[\frac{\alpha}{C}d_{ij}(\gamma_{ij}' - \gamma_{ij}'') + \frac{1-\alpha}{2}(|\gamma_{ij}' - \frac{1}{MN}| - |\gamma_{ij}'' - \frac{1}{MN}|)] \\
=& \sum_{i=1}^{M}\sum_{k=1}^{N}[\frac{\alpha}{C}d_{ij_k^i}(\gamma_{ij_k^i}' - \gamma_{ij_k^i}'') + \frac{1-\alpha}{2}(|\gamma_{ij_k^i}' - \frac{1}{MN}| - |\gamma_{ij_k^i}'' - \frac{1}{MN}|)] \\
=& \sum_{i=1}^{M}[\underbrace{\sum_{k=\hat{k}_i+1}^{N}\frac{\alpha}{C}(d_{ij_k^i} - d_{ij_1^i})\gamma_{ij_k^i}'}_{T_1} + \underbrace{\frac{1-\alpha}{2}\sum_{k=\hat{k}_i+1}^{N}(|\gamma_{ij_k^i}' - \frac{1}{MN}| - \frac{1}{MN})}_{T_2} + \\
& \underbrace{\frac{1-\alpha}{2}(|\gamma_{ij_1^i}' - \frac{1}{MN}| - |\gamma_{ij_1^i}' + \sum_{k=\hat{k}_i+1}^{N}\gamma_{ij_k^i}' - \frac{1}{MN}|)]}_{T_3}
\end{aligned}
$$

The last equation is due to the fact that $\gamma_{ij_k^i}'' = 0$ for $k > \hat{k}_i$ and $\gamma_{ij_1^i}'' = \gamma_{ij_1^i}' + \sum_{k=\hat{k}_i+1}^{N}\gamma_{ij_k^i}'$. Since $d_{ij_k^i} - d_{ij_1^i} \geq \frac{(1-\alpha)C}{\alpha}$ for any $k > \hat{k}_i$, we have $T_1 \geq (1-\alpha)\sum_{k=\hat{k}_i+1}^{N}\gamma_{ij_k^i}'$. By the triangle equality, we have $T_2 \geq \frac{1-\alpha}{2}\sum_{k=\hat{k}_i+1}^{N}(-\gamma_{ij_k^i}')$ and $T_3 \geq \frac{1-\alpha}{2}\sum_{k=\hat{k}_i+1}^{N}(-\gamma_{ij_k^i}')$. Therefore, we have $T_1 + T_2 + T_3 \geq 0$ and consequently $\mathcal{L}(\boldsymbol{\gamma}') \geq \mathcal{L}(\boldsymbol{\gamma}'')$.

Let $\mathcal{K}_{\text{high}}^i = \{2 \leq k \leq \hat{k}_i | \gamma_{ij_k^i}'' > \frac{1}{MN}\}$ and $\mathcal{K}_{\text{low}}^i = \{2 \leq k \leq \hat{k}_i | \gamma_{ij_k^i}'' < \frac{1}{MN}\}$. Let $\boldsymbol{\gamma}''' \in \mathbb{R}_{\geq 0}^{M\times N}$ be a probability transport such that

$$
\forall k \in \{2,\ldots,N\}, \gamma_{ij_k^i}''' = \begin{cases} \gamma_{ij_k^i}^*, & \text{if } k \in \mathcal{K}_{\text{high}}^i \\ \gamma_{ij_k^i}'', & \text{otherwise} \end{cases}
$$

Then we show that $\mathcal{L}(\boldsymbol{\gamma}'') \geq \mathcal{L}(\boldsymbol{\gamma}''')$. Since $\gamma_{ij_k^i}''' = \frac{1}{MN}$ for $k \in \mathcal{K}_{\text{high}}^i$ and $\gamma_{ij_1^i}''' = \gamma_{ij_1^i}'' + \sum_{k\in\mathcal{K}_{\text{high}}^i}(\gamma_{ij_k^i}'' - \frac{1}{MN})$, we have

$$
\begin{aligned}
\mathcal{L}(\boldsymbol{\gamma}'') - \mathcal{L}(\boldsymbol{\gamma}''') =& \sum_{i=1}^{M}[\underbrace{\sum_{k\in\mathcal{K}_{\text{high}}^i}\frac{\alpha}{C}(d_{ij_k^i} - d_{ij_1^i})(\gamma_{ij_k^i}'' - \frac{1}{MN})}_{T_4} + \underbrace{\frac{1-\alpha}{2}\sum_{k\in\mathcal{K}_{\text{high}}^i}|\gamma_{ij_k^i}'' - \frac{1}{MN}|}_{T_5} + \\
& \underbrace{\frac{1-\alpha}{2}(|\gamma_{ij_1^i}'' - \frac{1}{MN}| - |\gamma_{ij_1^i}'' + \sum_{k\in\mathcal{K}_{\text{high}}^i}(\gamma_{ij_k^i}'' - \frac{1}{MN}) - \frac{1}{MN}|)]}_{T_6}
\end{aligned}
$$

Again by the triangle inequality, we have $T_6 \geq -\frac{1-\alpha}{2}\sum_{k\in\mathcal{K}_{\text{high}}^i}|\gamma_{ij_k^i}'' - \frac{1}{MN}|$, and therefore $T_5 + T_6 \geq 0$. Since we also have $T_4 \geq 0$, it follows that $\mathcal{L}(\boldsymbol{\gamma}'') \geq \mathcal{L}(\boldsymbol{\gamma}''')$.

Finally, we show that $\mathcal{L}(\boldsymbol{\gamma}''') \geq \mathcal{L}(\boldsymbol{\gamma}^*)$. Since $\gamma_{ij_1^i}^* = \gamma_{ij_1^i}''' + \sum_{k\in\mathcal{K}_{\text{low}}^i}(\gamma_{ij_k^i}''' - \frac{1}{MN})$, we have

$$
\begin{aligned}
\mathcal{L}(\boldsymbol{\gamma}''') - \mathcal{L}(\boldsymbol{\gamma}^*) =& \sum_{i=1}^{M}[\underbrace{\sum_{k\in\mathcal{K}_{\text{low}}^i}\frac{\alpha}{C}(d_{ij_k^i} - d_{ij_1^i})(\gamma_{ij_k^i}''' - \frac{1}{MN})}_{T_7} + \underbrace{\frac{1-\alpha}{2}\sum_{k\in\mathcal{K}_{\text{low}}^i}|\gamma_{ij_k^i}''' - \frac{1}{MN}|}_{T_8} + \\
& \underbrace{\frac{1-\alpha}{2}(|\gamma_{ij_1^i}''' - \frac{1}{MN}| - |\gamma_{ij_1^i}''' + \sum_{k\in\mathcal{K}_{\text{low}}^i}(\gamma_{ij_k^i}''' - \frac{1}{MN}) - \frac{1}{MN}|)]}_{T_9}
\end{aligned}
$$

Since $d_{ij_k^i} - d_{ij_1^i} < \frac{(1-\alpha)C}{\alpha}$ for any $k \in \mathcal{K}^i_{\text{low}}$, we have $T_7 \geq (1-\alpha) \sum_{k \in \mathcal{K}^i_{\text{low}}} (\gamma'''_{ij_k^i} - \frac{1}{MN})$. Notice that $\gamma'''_{ij_1^i} \geq \frac{1}{MN}$ since $\gamma'''_{ij_1^i} = \frac{1}{M} - \sum_{k=2}^{N} \gamma'''_{ij_k^i}$ and for $k \in \{2, \ldots, N\}$, $\gamma'''_{ij_k^i} \leq \frac{1}{MN}$. We also have $\gamma'''_{ij_1^i} + \sum_{k \in \mathcal{K}^i_{\text{low}}} (\gamma'''_{ij_k^i} - \frac{1}{MN}) \geq \frac{1}{MN}$ since $\gamma^*_{ij_1^i} \geq \frac{1}{MN}$. Therefore, we have $T_8 + T_9 = (1-\alpha) \sum_{k \in \mathcal{K}^i_{\text{low}}} (\frac{1}{MN} - \gamma'''_{ij_k^i})$ and $T_7 + T_8 + T_9 \geq 0$. The it follows that $\mathcal{L}(\boldsymbol{\gamma}''') \geq \mathcal{L}(\boldsymbol{\gamma}^*)$.

Thus, we have $\mathcal{L}(\boldsymbol{\gamma}') \geq \mathcal{L}(\boldsymbol{\gamma}'') \geq \mathcal{L}(\boldsymbol{\gamma}''') \geq \mathcal{L}(\boldsymbol{\gamma}^*)$.

Next we show that if $\forall i \in [M] \forall k \in [N]$, $d_{ij_k^i} - d_{ij_1^i} \neq \frac{(1-\alpha)C}{\alpha}$ and $d_{ij_1^i} \neq d_{ij_2^i}$, $\boldsymbol{\gamma}^*$ is the unique solution. We consider two cases. In the first case where $\forall i \in [M] \forall k \in [N]$, $d_{ij_k^i} - d_{ij_1^i} < \frac{(1-\alpha)C}{\alpha}$, we have $\forall i \in [M] \forall j \in [N], \gamma^*_{ij} = \frac{1}{MN}$. For any $\boldsymbol{\gamma}' \neq \boldsymbol{\gamma}^*$, there must exist $i \in [M], k \in \{2, \ldots, N\}$ such that $\gamma'_{ij_k^i} > \frac{1}{MN}$ in which case $T_4 > 0$ or $\gamma'_{ij_k^i} < \frac{1}{MN}$ in which case $T_7 > (1-\alpha) \sum_{k \in \mathcal{K}^i_{\text{low}}} (\gamma'''_{ij_k^i} - \frac{1}{MN})$. In the second case where $\exists i \in [M] k \in [N]$ such that $d_{ij_k^i} - d_{ij_1^i} > \frac{(1-\alpha)C}{\alpha}$, we have $T_1 > (1-\alpha) \sum_{k=\hat{k}_i+1}^{N} \gamma'_{ij_k^i}$ for that $i$. In both cases, $\mathcal{L}(\boldsymbol{\gamma}') > \mathcal{L}(\boldsymbol{\gamma}^*)$ and thus $\boldsymbol{\gamma}^*$ is the unique solution.

$\square$

## B.2    Proof of Theorem 3.1 and Theorem 3.2

We show that Theorem 3.1 states a special case of Theorem 3.2. Then we prove Theorem 3.2 and it follows that Theorem 3.1 holds as well.

We first show the connection between Theorem 3.1 and Theorem 3.2. In Theorem 3.2, when $\rho_j = 1$ for all $j \in [N]$, $s_k^i = k$ and $s^*$ is the same as the $K$ in Theorem 3.1. Then the optimal solution in Theorem 3.2 is also the same as the one in Theorem 3.1 if we substitute $s^*$ by $K$ and all the $\rho_j$'s by 1.

Let $\mathcal{L}(\boldsymbol{\gamma}) = \frac{\alpha}{C} \sum_{i=1}^{M} \sum_{j=1}^{N} \gamma_{ij} d_{ij} + (1-\alpha) G_{\text{KDE}}(\boldsymbol{\gamma})$ be the optimization objective. We prove Theorem 3.2 by showing that for any $\boldsymbol{\gamma}' \in \mathbb{R}_{\geq 0}^{M \times N}$ that satisfy the constraint $(\forall i \in [M] \sum_{j=1}^{N} \gamma'_{ij} = \frac{1}{M})$, $\mathcal{L}(\boldsymbol{\gamma}') \geq \mathcal{L}(\boldsymbol{\gamma}^*)$.

For conciseness, we let $d_{(i,k)} = d_{ij_k^i}$ and $\gamma_{(i,k)} = \gamma_{ij_k^i}$.

We first show that $c(s)$ is a non-decreasing function. Since $c_i(s)$ is a step function and $\sum_{l=1}^{k} \frac{d_{(i,k+1)} - d_{(i,l)}}{\rho_{j_l^i}} - \sum_{l=1}^{k-1} \frac{d_{(i,k)} - d_{(i,l)}}{\rho_{j_l^i}} = \sum_{l=1}^{k} \frac{d_{(i,k+1)} - d_{(i,k)}}{\rho_{j_l^i}} \geq 0$ for any $k \in [N-1]$, $c_i(s)$ is non-decreasing. Therefore, $c_i(s) = \sum_{i=1}^{M} c_i(s)$ is non-decreasing.

Let $r' = \max_{i \in [M]} \max_{k \in [K_i]} \rho_{j_k^i} \gamma'_{(i,k)}$. We consider the following two cases.

In the first case when $r' \leq \frac{1}{Ms^*}$, we have

$$
\begin{aligned}
\mathcal{L}(\boldsymbol{\gamma}') - \mathcal{L}(\boldsymbol{\gamma}^*) = & \frac{\alpha}{C} \sum_{i=1}^{M} \sum_{k=1}^{N} d_{(i,k)} (\gamma'_{(i,k)} - \gamma^*_{(i,k)}) + (1-\alpha)(G_{\text{KDE}}(\boldsymbol{\gamma}') - G_{\text{KDE}}(\boldsymbol{\gamma}^*)) \\
= & \underbrace{\frac{\alpha}{C} \sum_{i=1}^{M} [\sum_{k=1}^{K_i} d_{(i,k)} (\gamma'_{(i,k)} - \gamma^*_{(i,k)}) + d_{(i,K_i+1)} (\gamma'_{(i,K_i+1)} - \gamma^*_{(i,K_i+1)}) + \sum_{k=K_i+2}^{N} d_{(i,k)} \gamma'_{(i,k)}]}_{T_1} \\
& \underbrace{(1-\alpha)(G_{\text{KDE}}(\boldsymbol{\gamma}') - G_{\text{KDE}}(\boldsymbol{\gamma}^*))}_{T_2}
\end{aligned}
$$

Since $\forall k \geq K_i + 2, d_{(i,k)} \geq d_{(i,K_i+1)}$, and $\sum_{k=K_i+1}^{N} \gamma'_{(i,k)} - \gamma^*_{(i,K_i+1)} = -\sum_{k=1}^{K_i}(\gamma'_{(i,k)} - \gamma^*_{(i,k)})$, we have

$$
\begin{aligned}
T_1 \geq & \frac{\alpha}{C} \sum_{i=1}^{M}\left[\sum_{k=1}^{K_i} d_{(i,k)}(\gamma'_{(i,k)} - \gamma^*_{(i,k)}) + d_{(i,K_i+1)}\left(\sum_{k=K_i+1}^{N} \gamma'_{(i,k)} - \gamma^*_{(i,K_i+1)}\right)\right] \\
= & \frac{\alpha}{C} \sum_{i=1}^{M} \sum_{k=1}^{K_i} \frac{d_{(i,K_i+1)} - d_{(i,k)}}{\rho_{j_k^i}} (\rho_{j_k^i} \gamma^*_{(i,k)} - \rho_{j_k^i} \gamma'_{(i,k)}) \\
\geq & \frac{\alpha}{C} \sum_{i=1}^{M} \sum_{k=1}^{K_i} \frac{d_{(i,K_i+1)} - d_{(i,k)}}{\rho_{j_k^i}}\left(\frac{1}{Ms^*} - r'\right)
\end{aligned}
$$

Let $\hat{s} = \min_{i \in [M]} s_{K_i+1}^i$. Then we have $\hat{s} > s^*$ and $\sum_{i=1}^{M} \sum_{k=1}^{K_i} \frac{d_{(i,K_i+1)} - d_{(i,k)}}{\rho_{j_k^i}} = c(\hat{s})$. Since $c(s)$ is non-decreasing, we have $\frac{\alpha}{C} c(\hat{s}) \geq (1-\alpha)M$. Then it follows that $T_1 \geq (1-\alpha)M\left(\frac{1}{Ms^*} - r'\right)$.

Let $\bar{s} = \sum_{j=1}^{N} 1/\rho_j$. Given the assumption that $s^* \leq \frac{1}{2}\bar{s}$, we have $\frac{1}{Ms^*} \geq 2\frac{1}{M\bar{s}}$. For any $i \in [M]$, for any $k \leq K_i$ we have $\rho_{j_k^i} \gamma^*_{(i,k)} = \frac{1}{Ms^*}$, and for $k = K_i+1$ we have $\rho_{j_k^i} \gamma^*_{(i,k)} = \frac{1}{Ms^*}(s^* - s_{K_i}^i)\rho_{j_k^i} \leq \frac{1}{Ms^*}(s_{K_i+1}^i - s_{K_i}^i)\rho_{j_k^i} = \frac{1}{Ms^*}$. Therefore, $\max_{i \in [M], k \in [N]} |\rho_{j_k^i} \gamma^*_{(i,k)} - \frac{1}{M\bar{s}}| = \frac{1}{Ms^*} - \frac{1}{M\bar{s}}$. Then we have

$$
\begin{aligned}
T_2 = & (1-\alpha)M\left(\max_{i \in [M], k \in [N]} |\rho_{j_k^i} \gamma'_{(i,k)} - \frac{1}{M\bar{s}}| - \max_{i \in [M], k \in [N]} |\rho_{j_k^i} \gamma^*_{(i,k)} - \frac{1}{M\bar{s}}|\right) \\
\geq & (1-\alpha)M\left(\max_{i \in [M]} \max_{k \in [K_i]} |\rho_{j_k^i} \gamma'_{(i,k)} - \frac{1}{M\bar{s}}| - \left(\frac{1}{Ms^*} - \frac{1}{M\bar{s}}\right)\right) \\
\geq & (1-\alpha)M\left(|r' - \frac{1}{M\bar{s}}| - \left(\frac{1}{Ms^*} - \frac{1}{M\bar{s}}\right)\right) \\
\geq & (1-\alpha)M\left(r' - \frac{1}{Ms^*}\right)
\end{aligned}
$$

The last inequality follows the triangle inequality. Then it follows that $T_1 + T_2 \geq 0$ and $\mathcal{L}(\gamma') - \mathcal{L}(\gamma^*) \geq 0$.

In the second case when $r' > \frac{1}{Ms^*}$, let $\hat{K}_i = \max\{K \in [N] \cup \{0\} | \sum_{k=1}^{K} r'/\rho_{j_k^i} \leq \frac{1}{M}\}$. When $K > K_i$, $\sum_{k=1}^{K} r'/\rho_{j_k^i} > s^* r' > \frac{1}{M}$. Therefore, $\hat{K}_i \leq K_i$. Consider another probability transport $\gamma'' \in \mathbb{R}_{\geq 0}^{M \times N}$ where

$$
\gamma''_{(i,k)} = \begin{cases} r'/\rho_{j_k^i}, & \text{if } k \leq \hat{K}_i \\ \frac{1}{M} - \sum_{k=1}^{\hat{K}_i} r'/\rho_{j_k^i}, & \text{if } k = \hat{K}_i + 1 \\ 0, & \text{otherwise} \end{cases}
$$

Note that by the definition of $\hat{K}_i$ we have $\gamma''_{(i,k)} \rho_{j_k^i} < r'$ for $k = \hat{K}_i + 1$.

Then we have

$$
\begin{aligned}
\mathcal{L}(\gamma') - \mathcal{L}(\gamma'') = & \frac{\alpha}{C} \sum_{i=1}^{M} \sum_{k=1}^{N} d_{(i,k)}(\gamma'_{(i,k)} - \gamma''_{(i,k)}) + (1-\alpha)(G_{\text{KDE}}(\gamma') - G_{\text{KDE}}(\gamma'')) \\
= & \underbrace{\frac{\alpha}{C} \sum_{i=1}^{M}\left[\sum_{k=1}^{\hat{K}_i} d_{(i,k)}(\gamma'_{(i,k)} - \gamma''_{(i,k)}) + d_{(i,\hat{K}_i+1)}(\gamma'_{(i,\hat{K}_i+1)} - \gamma''_{(i,\hat{K}_i+1)}) + \sum_{k=\hat{K}_i+2}^{N} d_{(i,k)}\gamma'_{(i,k)}\right]}_{T_3} \\
& \underbrace{(1-\alpha)(G_{\text{KDE}}(\gamma') - G_{\text{KDE}}(\gamma''))}_{T_4}
\end{aligned}
$$

Since $\forall k \geq \hat{K}_i + 2, d_{(i,k)} \geq d_{(i,\hat{K}_i+1)}$, and $\sum_{k=\hat{K}_i+1}^{N} \gamma'_{(i,k)} - \gamma''_{(i,\hat{K}_i+1)} = -\sum_{k=1}^{\hat{K}_i}(\gamma'_{(i,k)} - \gamma''_{(i,k)})$, we have

$$
\begin{aligned}
T_3 \geq &\frac{\alpha}{C} \sum_{i=1}^{M} [\sum_{k=1}^{\hat{K}_i} d_{(i,k)}(\gamma'_{(i,k)} - \gamma''_{(i,k)}) + d_{(i,\hat{K}_i+1)}(\sum_{k=\hat{K}_i+1}^{N} \gamma'_{(i,k)} - \gamma''_{(i,\hat{K}_i+1)})] \\
= &\frac{\alpha}{C} \sum_{i=1}^{M} \sum_{k=1}^{\hat{K}_i} \frac{d_{(i,\hat{K}_i+1)} - d_{(i,k)}}{\rho_{j_k^i}}(\rho_{j_k^i}\gamma''_{(i,k)} - \rho_{j_k^i}\gamma'_{(i,k)}) \\
\geq &0
\end{aligned}
$$

In addition, since $r' > \frac{1}{Ms^*} \geq \frac{2}{M\bar{s}}$ and $\rho_{j_k^i}\gamma''_{(i,k)} \leq r'$ for any $i \in [M]$ and $k \in [N]$, we have $\max_{i \in [M],k \in [N]} |\rho_{j_k^i}\gamma'_{(i,k)} - \frac{1}{M\bar{s}}| \geq \max_{i \in [M]} \max_{k \in [K_i]} |\rho_{j_k^i}\gamma'_{(i,k)} - \frac{1}{M\bar{s}}| = r' - \frac{1}{M\bar{s}}$ and $\max_{i \in [M],k \in [N]} |\rho_{j_k^i}\gamma''_{(i,k)} - \frac{1}{M\bar{s}}| \leq r' - \frac{1}{M\bar{s}}$. Therefore,

$$
\begin{aligned}
T_4 = &(1-\alpha)M(\max_{i \in [M],k \in [N]} |\rho_{j_k^i}\gamma'_{(i,k)} - \frac{1}{M\bar{s}}| - \max_{i \in [M],k \in [N]} |\rho_{j_k^i}\gamma''_{(i,k)} - \frac{1}{M\bar{s}}|) \\
\geq &0
\end{aligned}
$$

Then it follows that $\mathcal{L}(\gamma') - \mathcal{L}(\gamma'') \geq 0$

Let $\mathcal{S}' = \{s \in \mathcal{S}|\frac{1}{Mr'} < s \leq s^*\}$ and $s^{(1)}, \ldots, s^{(|\mathcal{S}'|)}$ be the elements in $\mathcal{S}'$ in the ascending order. Let $\gamma^{(0)} = \gamma''$, $s^{(0)} = \frac{1}{Mr'}$ and $K_i^{(0)} = \hat{K}_i$. For $t \in [|\mathcal{S}'|]$, let $K_i^{(t)} = \max\{k \in [N]|s_k^i \leq s^{(t)}\}$. we consider the probability transport $\gamma^{(t)} \in \mathbb{R}_{\geq 0}^{M \times N}$ where

$$
\gamma_{(i,k)}^{(t)} = \begin{cases} 1/\rho_{j_k^i} \cdot \frac{1}{Ms^{(t)}}, & \text{if } k \leq K_i^{(t)} \\ \frac{1}{M} - \sum_{k=1}^{K_i^{(t)}} 1/\rho_{j_k^i} \cdot \frac{1}{Ms^{(t)}}, & \text{if } k = K_i^{(t)} + 1 \\ 0, & \text{otherwise} \end{cases}
$$

Then we have

$$
\mathcal{L}(\gamma^{(t-1)}) - \mathcal{L}(\gamma^{(t)}) = \underbrace{\frac{\alpha}{C} \sum_{i=1}^{M} \sum_{k=1}^{N} d_{(i,k)}(\gamma_{(i,k)}^{(t-1)} - \gamma_{(i,k)}^{(t)})}_{T_5} + \underbrace{(1-\alpha)(G_{\text{KDE}}(\gamma^{(t-1)}) - G_{\text{KDE}}(\gamma^{(t)}))}_{T_6}
$$

By the definition of $K_i^{(t)}$ and $s^{(t)}$, either $K_i^{(t)} = K_i^{(t-1)}$ or $K_i^{(t)} = K_i^{(t-1)} + 1$. For any $i \in [M]$ such that $K_i^{(t)} = K_i^{(t-1)}$, we have $\gamma_{(i,k)}^{(t)} = 0$ for $k > K_i^{(t-1)} + 1$. For any $i \in [M]$ such that $K_i^{(t)} = K_i^{(t-1)} + 1$, we have $s_{K_i^{(t)}}^i = s^{(t)}$, in which case we also have $\gamma_{(i,k)}^{(t)} = 0$ for $k > K_i^{(t-1)} + 1$.

Therefore, we have $\gamma_{(i,K_i^{(t-1)}+1)}^{(t-1)} - \gamma_{(i,K_i^{(t-1)}+1)}^{(t)} = -\sum_{k=1}^{K_i^{(t-1)}}(\gamma_{(i,k)}^{(t-1)} - \gamma_{(i,k)}^{(t)})$. Then it follows that

$$
\begin{aligned}
T_5 = &\frac{\alpha}{C} \sum_{i=1}^{M} \sum_{k=1}^{K_i^{(t-1)}} (d_{(i,k)} - d_{(i,K_i^{(t-1)}+1)})(\gamma_{(i,k)}^{(t-1)} - \gamma_{(i,k)}^{(t)}) \\
= &\frac{\alpha}{C} \sum_{i=1}^{M} \sum_{k=1}^{K_i^{(t-1)}} (d_{(i,K_i^{(t-1)}+1)} - d_{(i,k)})/\rho_{j_k^i} \cdot \frac{1}{M} \cdot (\frac{1}{s^{(t)}} - \frac{1}{s^{(t-1)}})
\end{aligned}
$$

Let $\hat{s}^{(t)} = \min_{i \in [M]} s_{K_i^{(t-1)}+1}^i$, and then we have $T_5 = \frac{\alpha}{C} \cdot \frac{1}{M} \cdot (\frac{1}{s^{(t)}} - \frac{1}{s^{(t-1)}})c(\hat{s}^{(t)})$. Since $\hat{s}^{(t)} \leq s^*$ and $c(s)$ is non-decreasing, we have $\frac{\alpha}{C}c(\hat{s}^{(t)}) \leq \frac{\alpha}{C}c(s^*) < (1-\alpha)M$ and then it follows that $T_5 \geq (1-\alpha)(\frac{1}{s^{(t)}} - \frac{1}{s^{(t-1)}})$.

In addition, since $s^{(t-1)} < s^{(t)} \leq s^*$, we have $\rho_{j_k^i}\gamma^{(t-1)} > \rho_{j_k^i}\gamma^{(t)} \geq \frac{1}{Ms^*} \geq \frac{2}{M\bar{s}}$, and further

$$
\begin{aligned}
T_6 = &(1-\alpha)M(\max_{i \in [M],k \in [N]} |\rho_{j_k^i}\gamma_{(i,k)}^{(t-1)} - \frac{1}{M\bar{s}}| - \max_{i \in [M],k \in [N]} |\rho_{j_k^i}\gamma_{(i,k)}^{(t)} - \frac{1}{M\bar{s}}|) \\
= &(1-\alpha)(\frac{1}{s^{(t-1)}} - \frac{1}{s^{(t)}})
\end{aligned}
$$

Therefore, we have $\mathcal{L}(\boldsymbol{\gamma}^{(t-1)}) - \mathcal{L}(\boldsymbol{\gamma}^{(t)}) = T_5 + T_6 \geq 0$. Since $\boldsymbol{\gamma}' \geq \boldsymbol{\gamma}'' = \boldsymbol{\gamma}^{(1)} \geq \cdots \geq \boldsymbol{\gamma}^{(|\mathcal{S}'|)} = \boldsymbol{\gamma}^*$, we have $\boldsymbol{\gamma}' \geq \boldsymbol{\gamma}^*$.

If $\nexists s \in \mathcal{S}$ such that $\frac{\alpha}{C} c(s) = (1 - \alpha)M$ and $\nexists i \in [M]$ such that $d_{(i,K_i)} = d_{(i,K_i+1)}$ or $d_{(i,K_i+1)} = d_{(i,K_i+2)}$, we have $T_1 > (1 - \alpha)M(\frac{1}{Ms^*} - r')$ and $T_3 > 0$, and therefore $\boldsymbol{\gamma}' > \boldsymbol{\gamma}^*$, i.e., $\boldsymbol{\gamma}^*$ is the unique solution.

## C   Implementation Details

In this section, we provide details of the implementations.

**Implementation of Our Method**   For experiments in Section 5.2, GETKNN is implemented as two-stage retrieval. We first build a coarse Faiss [23] index for the data repository $D$ and use it to retrieve the 2000 nearest neighbors of each query example. The retrieved examples form a new set $D'$. Then we build a fine-grained index for $D'$ and use it to retrieve and return the 2000 nearest neighbors of each query example. COMPUTEKDE in KNN-KDE computes the kernel density of each example in $D'$ by retrieving its 1000 nearest neighbors using the fine-grained index. The coarse index is `OPQ56_112,IVF65536_HNSW32,PQ7+56`, and the fine-grained index is `IndexIVFFlat`. We refer the readers to the Faiss documentation[3] for the details of those indexes.

For experiments in Section 5.1, we use exact search for GETKNN to retrieve 5000 nearest neighbors of each query example and `IndexIVFFlat` for COMPUTEKDE.

**Encoding Process for Instruction Selection**   We encode the examples following [47] using rescaled and randomly projected gradients from a LLAMA-2-7B model finetuned on a random 5% of the data repository. Specifically, we finetune the base model on the randomly selected dataset for 4 epochs and use the gradients from the checkpoint at the end of each epoch as the example encoding. The dimension of the projected gradient from each epoch is 8,192. We refer the readers to [47] for more details. Then for each example, we multiply the gradients from the 4 checkpoints by the corresponding learning rate and concatenate them to get the final encoding, which is a 32,768-dimensional vector.

**Parameter Selection**   Note that our framework only has two effective parameters ($C$ is a constant to make sure that the transport cost and $G(\gamma)$ are on the same scale). The way we set the hyperparameters is as follows:

- We set $C$ to 5 when the embeddings are normalized.
- We set $h$ to the maximum distance between 10 hand-crafted near-duplicates. The intuition is that the points within the distance of $h$ will be considered as near-duplicates and the probability assigned to them will be reduced.
- $\alpha$ can be any value between 0.05 and 0.95, and the performance is not sensitive to it as long as it is not too small or too large (see Appendix E).

In practice, we can use a validation set and a small surrogate model to guide the parameter selection.

## D   Hyperparameters of Finetuning

We apply LoRA [20] for parameter-efficient instruction tuning for the experiments in Section 5.1. The hyperparameters are shown in Table 5. We use an NVIDIA A100 Tensor Core GPU with 40G memory for instruction tuning.

For the experiments in Section 5.2, the hyperparameters for continued pretraining are provided in Table 6 and those for supervised finetuning are in Table 7. The hardware for continued pretraining and supervised finetuning is an NVIDIA Tesla V100 GPU with 32GB memory.

## E   Additional Experimental Results

---

[3]https://github.com/facebookresearch/faiss

Table 5: Hyperparameters for instruction tuning.

| | |
|---|---|
| maximum token length | 2048 |
| batch size | 128 |
| epochs | 4 |
| optimizer | AdamW |
| weight decay | 0.0 |
| Adam $\beta_1$ | 0.9 |
| Adam $\beta_2$ | 0.999 |
| Adam $\epsilon$ | 1e-8 |
| warmup ratio | 0.03 |
| learning rate scheduler | cosine |
| learning rate | 2e-5 |
| LoRA rank | 128 |
| LoRA $\alpha$ | 512 |
| LoRA dropout rate | 0.1 |

Table 6: Hyperparameters for continued pretraining.

| | |
|---|---|
| maximum token length | 256 |
| batch size | 128 |
| optimizer | AdamW |
| weight decay | 0.01 |
| Adam $\beta_1$ | 0.9 |
| Adam $\beta_2$ | 0.999 |
| Adam $\epsilon$ | 1e-6 |
| warmup ratio | 0.1 |
| learning rate scheduler | linear |
| learning rate | 5e-4 |

Table 7: Hyperparameters for finetuning. We set patience for early stopping to 3 epochs so that finetuning stops when the validation F1 score does not increase for 3 epochs.

| | |
|---|---|
| maximum token length | 256 |
| batch size | 16 |
| epochs | 10 |
| patience for early stopping | 3 epochs |
| optimizer | AdamW |
| weight decay | 0.1 |
| Adam $\beta_1$ | 0.9 |
| Adam $\beta_2$ | 0.999 |
| Adam $\epsilon$ | 1e-6 |
| warmup ratio | 0.1 |
| learning rate scheduler | linear |
| learning rate | 5e-5 |

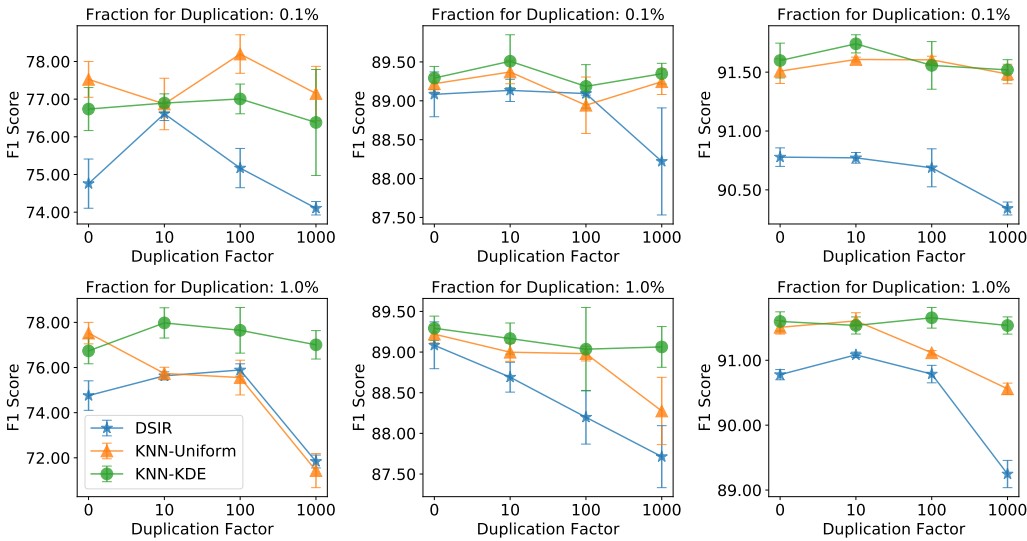

Figure 2: F1 scores of the downstream tasks under different duplication settings.

### E.1 Robustness to Near-Duplicates

We evaluate the robustness of the selection methods against near-duplicates in the candidate examples. We follow the same evaluation protocol described in Section 5.2 while injecting duplicates to the candidate examples. We set different levels of duplication by varying the fraction of examples chosen for duplication and the duplication factor (number of duplicates per example). The fraction for duplication is set to 0.1% / 1%, and the duplication factor is set to 10 / 100 / 1000. For example, if the fraction for duplication is 0.1% and the duplication factor is 10, we randomly choose 0.1% of the examples from the data repository and duplicate each 10 times. We use ChemProt (1K), AGNews (3K), and IMDB (10K) to perform the analysis, where the numbers in the parentheses represent the sizes of the annotated data. We include KNN-Uniform with the same parameters as KNN-KDE to show the effectiveness of the KDE-based regularization.

The results show that KNN-KDE is the only method that is robust to all the duplication settings. We observe that under low duplication levels, specifically when (fraction for duplication, duplication factor) is (0.1%, 10), (0.1%, 100), or (1%, 10), all the methods perform similarly to the case without duplication. Given that the injected duplicates constitute less than 10% of the data repository in those settings, it is not surprising that they do not have much effect on the downstream performance. However, when the duplication factor is increased to 1000 with the fraction for duplication set to 0.1%, the performance of DSIR drops by 0.7 points on average, whereas KNN-KDE and KNN-Uniform retain their performance. Moreover, when the duplication factor is increased to 1000 with the fraction set to 1%, all the methods except KNN-KDE show a notable decline (more than 2 points on average) in their performance.

### E.2 Runtime and Scalability

We report the runtime of our method that can be split into a pre-processing stage and a selection stage. We use a machine with an Intel(R) Xeon(R) Gold 5115 CPU @ 2.40GHz (40 cores) and 250GB RAM. The example embedding is computed using an NVIDIA Tesla V100 GPU with 32GB memory, while the other computations are on the CPU. In the pre-processing stage, our method embeds the candidate examples in the data repository and further builds indexes for the embeddings. This stage takes 28.38 hours for the data repository in Section 5.2 that contains 150M examples. In the selection stage, our method embeds the query examples, computes the probability assignment, and takes random samples according to the probability. This stage takes 0.7 hours for 10K query examples. The runtime of the selection stage scales linearly with the number of query examples and remains unaffected by the number of examples to be sampled except for the I/O cost. Note that while our method takes a substantial amount of time in the pre-processing stage, the cost is one-time and

the index can be reused for a variety of tasks that require similarity search. In general, our methods are practical in terms of runtime.

### E.3 Task-Specific Instruction Tuning for One epoch

In Section 5.1, we perform instruction tuning for 4 epochs, and our method takes a random sample in each epoch instead of using a fixed set. Therefore, the total number of unique examples can be up to 4x the number of examples used per epoch, though the actual number of unique examples is much lower since examples with high probability mass tend to be repeatedly sampled. To demonstrate that the number of unique examples during training is not the primary factor behind our performance gain, we provide additional results that compare our method with LESS when the number of epochs is set to 1. Specifically, each method selects a set whose size is 4% of the candidates. Then we train the model on the selected set for 1 epoch (the amount of computation is the same as using 1% for 4 epochs). The results are shown in Table 8. From the results, we can see that our method still outperforms LESS in 5 out of the 6 settings when LESS has access to more unique examples.

Table 8: Performance of instruction tuning with dataset selected by our method compared with the LESS. The dataset size is 4% of the candidate data repository and we train each model for one epoch on the selected set. The subscripts represent the standard deviations.

| Model | LLAMA-2-7B | | | MISTRAL-7B | | |
|---|---|---|---|---|---|---|
| Dataset | TydiQA | MMLU | BBH | TydiQA | MMLU | BBH |
| LESS | $54.4_{0.0}$ | $46.5_{0.9}$ | $40.4_{1.3}$ | $60.5_{1.6}$ | $\mathbf{60.8_{0.4}}$ | $55.8_{1.5}$ |
| Ours | $\mathbf{55.4_{0.5}}$ | $\mathbf{47.9_{0.2}}$ | $\mathbf{42.0_{1.1}}$ | $\mathbf{63.6_{1.4}}$ | $60.5_{0.8}$ | $\mathbf{56.3_{2.1}}$ |

### E.4 Domain-Specific Continued Pretraining with Different Selection Sizes

We compare with the baselines when the size of the selected data is 100K and 300K for domain-specific continued pretraining while the size of the annotated dataset is fixed to 3K. The other settings are the same as in Section 5.2. The results are in Table 9 which show that our method is either better than or comparable to the baselines.

Table 9: F1 scores of the downstream tasks when the sample size varies. The size of the annotated data is set to 3K. Standard deviations are shown in the subscripts.

| | ——100K Sample Size —— | | | | ——300K Sample Size —— | | | |
|---|---|---|---|---|---|---|---|---|
| | ChemP. | IMDB | SCI. | AGNews | ChemP. | IMDB | SCI. | AGNews |
| Base | $77.1_{1.1}$ | $88.7_{0.4}$ | $75.8_{1.1}$ | $87.7_{0.3}$ | $77.1_{1.1}$ | $88.7_{0.4}$ | $75.8_{1.1}$ | $87.7_{0.3}$ |
| Rand | $77.8_{0.4}$ | $88.9_{0.2}$ | $78.7_{0.9}$ | $88.5_{0.3}$ | $78.2_{0.3}$ | $89.2_{0.2}$ | $78.8_{0.2}$ | $88.5_{0.2}$ |
| DSIR | $\mathbf{80.9_{0.9}}$ | $89.0_{0.3}$ | $79.9_{1.0}$ | $89.0_{0.1}$ | $\mathbf{82.1_{0.3}}$ | $89.4_{0.3}$ | $78.2_{0.4}$ | $88.9_{0.2}$ |
| Ours | $80.4_{0.8}$ | $\mathbf{90.1_{0.1}}$ | $\mathbf{80.5_{0.4}}$ | $\mathbf{89.3_{0.1}}$ | $81.6_{0.1}$ | $\mathbf{90.2_{0.3}}$ | $\mathbf{79.8_{0.2}}$ | $\mathbf{89.2_{0.1}}$ |

### E.5 Micro-Benchmarks

In this section, we provide micro-benchmarks that study the effects of the hyperparameters in our framework. In addition, we show the performance of KNN-TV, an instantiation of our framework that is not covered in the main experiments. The experiments focus on domain-specific pretraining, following the same settings as in Section 5.2. The datasets and sizes used in the micro-benchmarks are ChemP (1K), AG (3K), and IMDB (10K).

#### E.5.1 Tradeoff between Distribution Alignment and Diversity

We study the effects of $\alpha$, the hyperparameter that controls the tradeoff between distribution alignment and diversity in our framework. We vary the value of $\alpha$ in KNN-Uniform and KNN-KDE and report the F1 scores of the downstream tasks in Figure 3. In all three datasets, we observe a notable drop in F1 scores when $\alpha = 0$ or $\alpha = 1$, and consistent performance when the value of $\alpha$ is set to other

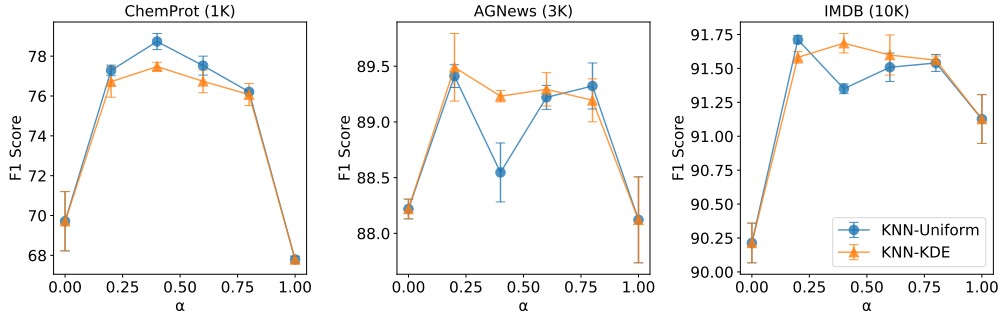

Figure 3: Performance of KNN-KDE when $\alpha$ varies. The error bar shows the standard deviation.

Table 10: Performance of KNN-KDE when the kernel size varies. F1 scores of the downstream tasks are reported with standard deviations shown in the subscripts.

| Kernel Size | 0.1 | 0.3 | 0.5 |
|---|---|---|---|
| ChemProt (1K) | $76.7_{0.5}$ | $76.8_{1.1}$ | $78.0_{0.3}$ |
| AGNews (3K) | $89.2_{0.1}$ | $89.3_{0.1}$ | $89.2_{0.2}$ |
| IMDB (10K) | $91.5_{0.1}$ | $91.4_{0.2}$ | $91.8_{0.0}$ |

values. Note that KNN-Uniform or KNN-KDE is equivalent to Uniform when $\alpha = 0$, and transports all the probability mass of each query example to its 1-nearest-neighbor in the data repository when $\alpha = 1$. The former does not consider distribution alignment, while the latter results in overfitting to the 1-nearest-neighbors. For the other values of $\alpha$, we report the corresponding neighborhood size (the final $K$ in KNN-Uniform and the average of the final $K_i$ in KNN-KDE) in Table 12. The consistent performance with $\alpha \in \{0.2, 0.4, 0.6, 0.8\}$ shows that our framework is not sensitive to the choice of $\alpha$.

### E.5.2 Effects of Kernel Size in KNN-KDE

We vary the kernel size for the kernel density estimation in KNN-KDE. The performance is shown in Table 10. The F1 scores of all three downstream tasks are consistent across different choices of kernel size. The results show that the performance of KNN-KDE is not sensitive to the choice of kernel size.

### E.5.3 Performance of KNN-TV

We evaluate KNN-TV ($C = 0.25$, $\alpha = 0.6$) and show the results in Table 11. KNN-TV performs similarly to KNN-KDE ($\alpha = 1$), and significantly worse than KNN-KDE ($\alpha = 0.6$). The reason is that KNN-TV assigns almost all the probability mass (more than 99.99%) to the 1-nearest neighbor of each query example and causes overfitting to them, a behavior similar to KNN-KDE ($\alpha = 1$).

Table 11: The performance of KNN-TV compared with KNN-KDE ($\alpha = 1$) and KNN-KDE ($\alpha = 0.6$). F1 scores of the downstream tasks are reported with standard deviations shown in the subscripts.

| Dataset | ChemProt (1K) | AGNews (3K) | IMDB (10K) |
|---|---|---|---|
| KNN-TV | $65.8_{1.6}$ | $88.0_{0.7}$ | $91.0_{0.1}$ |
| KNN-KDE ($\alpha = 1$) | $67.7_{0.1}$ | $88.1_{0.3}$ | $91.1_{0.1}$ |
| KNN-KDE ($\alpha = 0.6$) | $76.7_{0.5}$ | $89.2_{0.1}$ | $91.5_{0.1}$ |

Table 12: The neighborhood size of KNN-Uniform / KNN-KDE for different values of $\alpha$. The numbers before the slashes are for KNN-Uniform and those after are for KNN-KDE.

| Dataset | ChemProt (1K) | AGNews (3K) | IMDB (10K) |
|---|---|---|---|
| $\alpha = 0.2$ | 959 / 993 | 813 / 830 | 918 / 928 |
| $\alpha = 0.4$ | 398 / 408 | 342 / 346 | 372 / 373 |
| $\alpha = 0.6$ | 189 / 191 | 165 / 164 | 177 / 174 |
| $\alpha = 0.8$ | 76 / 74 | 69 / 65 | 72 / 68 |

