# OpenReview forum: "TSDS: Data Selection for Task-Specific Model Finetuning"
_NeurIPS.cc/2024/Conference — NeurIPS 2024 poster_

### Official Review · Reviewer_CE8N · 2024-07-12

**Soundness:** 2
**Presentation:** 3
**Contribution:** 2
**Rating:** 3
**Confidence:** 5

**Summary:**

This paper proposes a method for data selection in foundation model fine-tuning. The proposal contains a distribution alignment loss based on optimal transport to capture the discrepancy between the selected data and the target distribution, a regularizer to encourage the diversity of the selected data, and kernel density estimation to reduce the negative effects of near-duplicates among the candidate data. Experimental on fine-tuning the language model are reported.

**Strengths:**

1. The proposal studied in this paper is interesting since how to select data, and how to improve the data quality is important for the training and fine-tuning of the foundation model.
2. The proposed method which considers distribution discrepancy minimization, diversity, and near-duplicates, is technically sound.

**Weaknesses:**

1. The novelty and contribution of the proposed method are limited. Data selection is important and well-studied in the machine learning community. For example, in active learning, we need to select examples to label according to some metrics; in domain adaptation, we need to select data to help the model reuse. Some widely adopted methods can be applied to the problem of data selection for the foundation model and the authors didn't provide a comprehensive study and comparison. Moreover, the techniques adopted in the proposal are also widely used techniques.

2. For the experiments, the authors only conduct experiments on the language model, can the proposal be applied to other foundation models, such as the vision-language model?

3. It seems that the random selection method can also achieve good performance. So I am wondering about the difficulty of the problem,  maybe we can improve the performance using some trivial techniques.

4. The time cost between fine-tuning with the full dataset and the selected dataset should be reported.

**Questions:**

As discussed in the Weakness part.

**Limitations:**

Yes

---

> ### Author Rebuttal · Authors · 2024-08-07
>
> We believe that you have missed key points of our work and we would like to correct several factually incorrect points in the provided review.
>
> > W1 - novelty
>
> To the best of our knowledge, our work is the first one that presents a unified framework for task-specific data selection which considers distribution matching, diversification, and deduplication. In addition, our formulation is general, supporting any metric spaces and distance functions where efficient nearest-neighbor search algorithms exist.
>
> > W2 - extension to VLMs
>
> We disagree that the focus on language models poses a weakness of our work. For task-specific data selection, it is common practice to use large language models for the evaluation [1][2]. In addition, we already show the effectiveness of our framework in diverse experimental settings. Evaluation using vision-language models is an interesting extension that can be studied in future works.
>
> >W3 - performance of random selection
>
> This statement is factually incorrect. Our method is significantly better than random selection with large gaps. In Table 2, we can see that our method consistently outperforms random selection, with a gap of 3.2 F1 points on average for task-specific finetuning. In Table 4, we show that our method consistently outperforms random selection, with a gap of 2.9 F1 points on average for domain-specific continued pretraining.
>
> >W4 - finetuning cost
>
> Finetuning on the full dataset takes 80 hours on an A100 GPU with 40G memory. The cost of finetuning on the selected set is based on the selection ratio. For example, if the selection ratio is 1%, the finetuning time is 80 * 1% = 0.8 hours. We will add the details to the Appendix.
>
> [1] Xia, Mengzhou, et al. "Less: Selecting influential data for targeted instruction tuning." International Conference on Machine Learning. PMLR, 2024.
>
> [2] Xie, Sang Michael, et al. "Data selection for language models via importance resampling." Advances in Neural Information Processing Systems 36 (2023): 34201-34227.

---

### Official Review · Reviewer_rjVY · 2024-07-12

**Soundness:** 2
**Presentation:** 2
**Contribution:** 3
**Rating:** 5
**Confidence:** 3

**Summary:**

This paper formulates data selection for task-specific fine-tuning as an optimization problem based on optimal transport for distribution alignment. It proposes two KNN-based implementation methods and evaluates them on datasets for task-specific instruction fine-tuning and domain-specific continued pretraining. The experimental results demonstrate that their methods are more effective than the baseline systems (LESS and DSIR).

**Strengths:**

1. The paper formulates data selection as an optimal transport problem, providing a detailed problem definition and a closed-form solution. Additionally, it proposes using Kernel Density Estimation to address the issue of near-duplicates.

2. The authors introduce KNN-Uniform and KNN-KDE algorithms for data selection, showing that their performance is superior to the baseline systems in both task-specific instruction fine-tuning and domain-specific continued pretraining experimental setups.

**Weaknesses:**

**Regarding the methodology:**

1. The connection between data selection and the optimal transport problem is not clearly established. Despite mentioning it in lines 114-115 of Section 3, it remains unclear why data selection can be considered an optimal transport problem.

2. Much of the paper is based on LESS, including the representation of samples and the task definition. However, there is minimal mention of LESS, making it challenging to understand without prior knowledge of LESS.

3. The method still relies on M query samples for a specific task, which poses certain limitations.

**Regarding the experimental section:**

4. The experimental section contains too many specific settings. For instance, special settings mentioned in lines 248 and 286 make it difficult to determine how these parameters were chosen, even after reviewing the appendix.

5. The two experimental setups are inconsistent in task-specific instruction fine-tuning and domain-specific continued pretraining. In Table 2, the Ratio of 0.5%-5% can be understood as the number of data samples selected. However, in Table 4, the 1K, 3K, and 10K seem to refer to the number of query samples, but there is no comparison of the number of selected samples.

6. There is a lack of comparison with various baseline systems. Only one baseline system is used for comparison, and its performance differs from that reported in the original paper.

**Questions:**

See weakness.

**Limitations:**

They acknowledge one limitation in conclusion: the method still relies on M query samples for a specific task. It is also raised in my comments.

---

> ### Author Rebuttal · Authors · 2024-08-07
>
> Thank you for your feedback. We address the concerns and answer the questions below:
>
> >W1 - connection to optimal transport
>
> As mentioned in lines 38-44, we want the selected data to match the distribution of the representative data from the target distribution, and optimal transport is a powerful tool to measure distribution alignment. We will highlight this connection in Section 3.1 when we introduce the optimization problem.
>
> >W2 - discussion of LESS
>
> We will add more discussion on LESS to related works. Specifically,  LESS computes the gradient similarity between each candidate and all the query examples, and the maximum similarity is the score for ranking. Then the top-ranked candidates are selected. A major difference between our method and LESS is that our method matches the distributions, while LESS takes the top ones based on aggregated statistics which can make it focus on a particular set of query examples.
>
> >W3 - need M query examples
>
> In practice, M can be a budget limit on how much human effort a user is willing to put into the selection process.  We will expand our discussion in the conclusion section to include this point.
>
> >W4 - how to set parameters
>
> Note that we only have two effective parameters ($C$ is a constant to make sure that the transport cost and $G(\gamma)$ are on the same scale). The way we set the hyperparameters is as follows:
> - We set $C$ to 5 when the embeddings are normalized.
> - We set $h$ to the maximum distance between 10 hand-crafted near-duplicates. The intuition is that the points within the distance of h will be considered as near-duplicates and the probability assigned to them will be reduced.
> - $\alpha$ can be any value between 0.05 - 0.95, and the performance is not sensitive to it as long as it is not too small or too large (see Appendix E.3.1). In practice, we can also use a validation set and a small surrogate model to guide the parameter selection.
>
> We will add the details to the appendix.
>
> >W5 - evaluation using different sample sizes for continued pretraining
>
> We did experiments that selected 100K and 300K examples for domain-specific continued pretraining  (see Table 2 in the PDF attached to the global response). The observation is similar to the 1M sample size: our method is either better than or comparable to the baseline. We will add this result to the appendix.
>
> >W6 - baseline systems
>
> We compare with the current state-of-the-art for each setting. The discrepancy between our reported numbers and the numbers in the LESS paper is due to the updates to open-instruct. We followed LESS to use open-instruct for the evaluation, and we used the updated version. In fact, we discussed this finding with the authors of LESS and confirmed the discrepancy reasons using their old open-instruct scripts. We will make the evaluation script public so that people can reproduce our numbers.

---

> > ### Comment · Reviewer_rjVY · 2024-08-14
> > **Reply by the Reviewer**
> >
> > Thank you for your reply. I have read your response and other reviewers' comments. I will keep my score.

---

### Official Review · Reviewer_zKiE · 2024-07-13

**Soundness:** 4
**Presentation:** 4
**Contribution:** 3
**Rating:** 8
**Confidence:** 2

**Summary:**

This paper presents a method for data selection for task-specific model finetuning. The method relies on a small, representative sample of data from the target task to select matching, relevant data from a corresponding corpus. The method relies on framing this task as an optimization problem, utilizing an optimal-transport-based distribution alignment loss and a KDE-based regularizer to avoid oversampling (near-)duplicates.
The authors show this method to be highly scalable and data efficient, being competitive with, and often outperforming state-of-the-art methods for domain-specific continued pretraining and task-specific instruction tuning.

**Strengths:**

- The paper rigorously presents and tests the proposed method, with a detailed theoretical motivation.
- Sections 2-4 are well structured and introduce the method in a clear, progressive way.
- Performance results, especially for very small sample sizes, strongly support the utility of this method.

**Weaknesses:**

- Section 5.1: Given how different some of the performances are between llama and mistral, including other LLMs may give a more complete picture of the efficacy of this method.
- Section 5: Efficiency claims would benefit from context. How does the 28 hour initialization time compare to other SOTA methods on this dataset? How does it scale after initialization is done when repeatedly drawing task-specific samples compared to other methods?

**Questions:**

Again, comparing efficiency with other SOTA methods on the datasets used in this paper would be helpful in better contextualizing the performance presented.

**Limitations:**

The authors address limitations.

---

> ### Author Rebuttal · Authors · 2024-08-07
>
> We appreciate the feedback. Here is our response to the questions and concerns:
>
> >W1 - other LLMs
>
> Thank you for your suggestions. We chose llama-7b and mistral-7b since they achieved state-of-the-art performance in various tasks at the time of submission among the 7b-size models and are shown to be effective with LoRA finetuning. Our experiments in Section 5.1 demonstrate that the benefits of our data selection solution hold across both models. A detailed study of other models could be explored in a future, extended version of this work.
>
> > W2 & Q1 - runtime
>
> On the same 150M examples (used in the experiment in Section 5.2), DSIR takes 18 hours to initialize. Note that this is a one-time cost for the data repository and the index we build can be reused across tasks. Although our method requires an additional 10 hours to initialize compared to DSIR, it achieves an average gain of 1.92 F1 points when provided with 1K representative examples from the target task.
> For the task-specific selection after initialization, our method takes 0.11 hours and DSIR takes 0.13 hours when taking 1M examples guided by 10K query examples. The task-specific selection after initialization scales linearly with respect to the number of query examples, and is not affected by the number of examples we want to sample (except for the IO cost).
> For the experiment in Section 5.1, the initialization time for both LESS and our method is 54 hours, dominated by the gradient computation time for the 270K candidates. For each task-specific selection, both take less than 1 minute.
> We will add the discussion above to the appendix.

---

### Official Review · Reviewer_mg1e · 2024-07-13

**Soundness:** 3
**Presentation:** 3
**Contribution:** 3
**Rating:** 5
**Confidence:** 3

**Summary:**

This paper proposes task-specific training data selection for language model fine-tuning. Given a (small) set of representative examples for a task and a large set $D$ of possible training examples, the proposed method uses (regularized) optimal transport to assign a probability distribution over $D$ that matches the distribution of representative examples while also encouraging diversity among the elements of $D$ assigned a nonzero probability.
The authors prove that with a certain choice of regularization function, this is equivalent to (an adaptive version of) $k$-nearest neighbor selection of candidate data similar to the representative examples. Since $k$NN treats near-duplicates as distinct examples (which would decrease diversity of the selected data), the paper additionally introduces another regularization term based on kernel density estimation; the optimal transport with this regularization is a weighted $k$NN that intuitively accounts for the frequency of near-duplicates for each example.

**Strengths:**

- Good data selection is an important problem given that today's models are both expensive to fine-tune and very sample-efficient *if* they are given the "correct" high-quality fine-tuning data [1]. Most high-performing efforts still tweak the composition of these small task-specific datasets by hand. This paper has an interesting new take on framing task-specific data selection as an optimal transport problem between representative task examples and a large pool of candidate training data.

- Theorems 3.1 and 3.2 shows that with certain regularization terms, the optimal transport selection procedure is equivalent to certain variations of $k$-nearest-neighbor. This allows for efficient computation of the optimal data selection under this objective.

- The proposed approach can naturally be combined with approximate nearest neighbor search methods for efficiency.

- Strong empirical results showing that the proposed selection procedure can even outperform tuning with the full candidate dataset.

- The experiments include standard deviations across three runs, giving a sense of how big the gains are compared to noise.

[1] Zhou, Chunting, Pengfei Liu, Puxin Xu, Srini Iyer, Jiao Sun, Yuning Mao, Xuezhe Ma et al. "LIMA: less is more for alignment." In Proceedings of the 37th International Conference on Neural Information Processing Systems, pp. 55006-55021. 2023.

**Weaknesses:**

- Missing an ablation using embeddings instead of gradients, or any other distance function for the examples.

- Missing several relevant data selection baselines that also encourage diversity, e.g. Deita [1], QDIT [2], and methods based on DPPs [3].

- Changing the data mix changes the optimal learning rate (e.g., since it changes the scale of the loss function at initialization). The paper compares models trained on different data mixes with the same learning rate, but the fair comparison is optimal : optimal. It's not clear from the experiments whether the reported gains are due to the learning rate being more optimal for the selected data mix, especially since the metric used to select the data is based on the gradients of a model.

[1] Liu, W., Zeng, W., He, K., Jiang, Y., & He, J. What Makes Good Data for Alignment? A Comprehensive Study of Automatic Data Selection in Instruction Tuning. In The Twelfth International Conference on Learning Representations.

[2] Bukharin, Alexander, and Tuo Zhao. "Data diversity matters for robust instruction tuning." arXiv preprint arXiv:2311.14736 (2023).

[3] Wang, P., Shen, Y., Guo, Z., Stallone, M., Kim, Y., Golland, P., & Panda, R. (2024). Diversity Measurement and Subset Selection for Instruction Tuning Datasets. arXiv preprint arXiv:2402.02318.

**Questions:**

- It might be helpful to have some (simple) intuition after L140 explaining why regularizing distance to the uniform transport encourages diversity.

- If the optimal transport formulation is equivalent to a certain type of $k$NN, why not just present the method as a type of $k$NN? $k$NN has a long history in data selection going back to at least Wilson (1972). It's not clear what the optimal transport discussion buys other than added complexity.

- Given the optimal transport framing, I think there should be some discussion of other (efficient) regularized optimal transport algorithms, such as Sinkhorn? [2]

- If I understand L252--L254 correctly, the effective dataset size for the proposed method is actually up to 4x the reported size, because the data are resampled from the computed distribution each epoch. Does LESS (the baseline) get the same advantage? I.e., does LESS get to use 4x the data or do some kind of resampling?

[1] Wilson, Dennis L. "Asymptotic properties of nearest neighbor rules using edited data." IEEE Transactions on Systems, Man, and Cybernetics 3 (1972): 408-421.'

[2] Cuturi, Marco. "Sinkhorn distances: Lightspeed computation of optimal transport." Advances in neural information processing systems 26 (2013).

**Limitations:**

The paper includes a reasonable discussion of its limitations, but it could be improved by discussing some of the additional limitations with the empirical results mentioned above.

---

> ### Author Rebuttal · Authors · 2024-08-07
>
> Thank you for the feedback. We address the concerns and clarify the questions as follows:
>
> >W1 - ablation study using embeddings
>
> We proposed a framework where the data can be embedded in any metric space with any distance function that supports efficient nearest neighbor search. Finding the best choice of embedding and distance is orthogonal to our study. Our choice of using gradients for task-specific finetuning is motivated by LESS [1], a state-of-the-art method for task-specific finetuning at the time of submission, which shows that similarities of gradients are better approximations of influence on task-specific performance than similarities of embeddings. Based on their observations, we further show that applying our framework using exactly the same encoding and distance function yields even better results due to our distribution matching and diversification. For the experiments on domain-specific continued pretraining, we use sentence embeddings and also show strong results.
>
> >W2 - additional methods
>
> Thanks for the pointers. These methods target different settings in the broader data selection space and we find them not directly comparable to the setting we consider, i.e., task-specific data selection where not all candidates are relevant. We will add a discussion on these techniques in related works, as their diversification methods can be of interest to the reader.
>
> >W3 - hyperparameter setting
>
> We follow the convention of the data selection literature [1, 2, 3] to use the same hyperparameters in the evaluation. We use the hyperparameters recommended by previous works (see the experimental settings in Section 5), and they are not tailored to our selected data. Given that our methods achieve consistently good performance, it is unlikely that our method outperforms the others due to a more favorable set of hyperparameters.
>
> >Q1 - intuition behind uniform transport
>
> Thank you for the suggestion. We will add a discussion after L140 to highlight that uniform transport represents the most diverse case where every example receives the same amount of probability mass. Therefore, we use it as a reference point to encourage the distribution to be close to it. We can gain more intuition from Theorem 3.1, where we will add a discussion stating that as we increase the weight of the regularizer, the optimal K also increases.
>
> >Q2 - why use optimal transport instead of presenting the method as knn
>
> The key innovation of our framework goes beyond the kNN algorithms themselves. Our framework considers two essential objectives in data selection: distribution matching and diversification [4]. By formulating an optimization problem, we integrate these goals while also taking near-duplicates into account. The formulation also allows us to study the optimality of the problem and have optimal solutions (Theorem 3.1 and 3.2). In our framework, kNN serves as a tool for finding the optimal solutions.
>
> >Q3 - other efficient optimal transport algorithms
>
> Thanks for the suggestion. Incorporating algorithms like Sinkhorn for even further efficient computation is an interesting direction. We will add a discussion to related works.
>
> >Q4 - effective dataset size
>
> - We sample from the computed distribution with replacement every step. Therefore, the total number of unique examples can be up to 4x the number of examples used per epoch, though the actual number of unique examples is much lower since examples with high probability mass tend to be repeatedly sampled.
> LESS is a deterministic selection method, and we follow the setting in their paper to select a fixed set and use the selected set for 4 epochs. Then for fair comparison, when evaluating our method, we train for the same number of steps with the same batch size.
> - It's important to note that having more unique examples does not necessarily lead to better results, since many candidate examples may not be relevant to the task. For instance, the "Full" method, which has seen all candidate examples, performs worse than both LESS and our method in most cases.
> - To demonstrate that the number of unique examples during training is not the primary factor behind our performance gain, we provide additional results that compare our method with LESS when the number of epochs is set to 1 (see Table 1 in the PDF attached to the global response). Specifically, each method selects a set whose size is 4% of the candidates. Then we train the model on the selected set for 1 epoch  (the amount of computation is the same as using 1% for 4 epochs). From the results, we can see that our method still outperforms LESS in 5 out of the 6 settings when LESS has a larger “effective dataset size”.
>
> We will add a discussion with the additional experiment to the appendix.
>
> [1] Xia, Mengzhou, et al. "Less: Selecting influential data for targeted instruction tuning." International Conference on Machine Learning. PMLR, 2024.
>
> [2] Xie, Sang Michael, et al. "Data selection for language models via importance resampling." Advances in Neural Information Processing Systems 36 (2023): 34201-34227.
>
> [3] Coleman, Cody, et al. "Selection via proxy: Efficient data selection for deep learning." International Conference on Learning Representations (2019).
>
> [4] Albalak, Alon, et al. "A survey on data selection for language models." arXiv preprint arXiv:2402.16827 (2024).

---

### Author Rebuttal · Authors · 2024-08-07

Following the reviews, we would like to expand on our choice of using optimal transport as a means to solve the problem of task-specific selection. We use optimal transport to capture the discrepancy between the distribution we will sample from and the target distribution. We include probability transport cost in the optimization objective to encourage alignment between the two distributions.
We formulate our framework as an optimization problem to provide guidance for task-specific data selection. By framing the problem this way, we are able to integrate and balance two crucial objectives of task-specific data selection: distribution matching and diversification.
We find this new formulation also allows us to obtain practical solutions that achieve state-of-the-art results for a diverse array of settings. We will update our introduction to clarify the above regarding the novelty of our work.

---

### Decision · Program_Chairs · 2024-09-25

**Decision:**

Accept (poster)

**Comment:**

Summary:
The paper presents a method for selecting data for fine-tuning LLMs. Starting with a small representative set of examples for fine-tuning, the method formulates the data selection problem as an optimization problem with a distribution alignment loss from optimal transport to capture the discrepancy between the selected data and the target distribution. Empirical evidence shows that the method is scalable, data efficient, and competitive or better than the state-of-the-art baselines for this task.

Strengths:
* The paper presents a novel technique for task-specific data selection for finetuning LLMs, an important problem in machine learning.
* The proposed approach is theoretically well-grounded and motivated
* The empirical evidence strongly supports the effectiveness of the proposed approach
* The experimental validation is quite rigorous
* The paper is well-written and easy to understand

Weaknesses
* Discussion and comparison (wherever possible) with the previously proposed data selection methods is lagging and could be improved.
* Given the huge difference in performance between the two LLMs testing, the paper would benefit from showing the performance across some more open-source LLMs

Overall Recommendation:
The paper presents a novel approach based on optimal transport to select data efficiently for fine-tuning LLMs on a specific task. The approach is well-motivated, the experimental results are convincing, and the paper is clearly written. The overall recommendation is to accept this paper for publication, subject to the proposed revisions, as it will benefit the wider ML community.